# A General Framework for Inference-Time Scaling and Steering of Diffusion Models

**Raghav Singhal** [* 1]  **Zachary Horvitz** [* 2]  **Ryan Teehan** [* 1]
**Mengye Ren** [1]  **Zhou Yu** [2 3]  **Kathleen McKeown** [2]  **Rajesh Ranganath** [1 4]

## Abstract

Diffusion models have demonstrated remarkable performance in generative modeling, but generating samples with specific desiderata remains challenging. Existing solutions — such as fine-tuning, best-of-n sampling, and gradient-based guidance — are expensive, inefficient, or limited in applicability. In this work, we introduce Feynman-Kac (FK) steering, which applies Feynman-Kac interacting particle systems to the inference-time steering of diffusion models with arbitrary reward functions. FK steering works by generating multiple trajectories, called *particles*, and resampling particles at intermediate steps based on scores computed using functions called *potentials*. Potentials are defined using rewards for intermediate states and are chosen such that a high score indicates the particle will yield a high-reward sample. We explore various choices of potentials, rewards, and samplers. Steering text-to-image models with a human preference reward, we find that FK steering outperforms fine-tuned models with just 2 particles. Moreover, FK steering a 0.8B parameter model outperforms a 2.6B model, achieving state-of-the-art performance on prompt fidelity. We also steer text diffusion models with rewards for text quality and rare attributes such as toxicity, and find that FK steering generates lower perplexity text and enables gradient-free control. Overall, inference-time scaling and steering of diffusion models, even training-free, provides significant quality and controllability benefits. Code available here.

---

[*]Equal contribution [1]Courant Institute of Mathematical Sciences, New York University [2]Columbia University [3]Arxlex.ai [4]Center for Data Science, New York University. Correspondence to: Raghav Singhal <rsinghal@nyu.edu>, Zachary Horvitz <zfh2000@columbia.edu>, Ryan Teehan <rst306@nyu.edu>.

*Proceedings of the 42$^{nd}$ International Conference on Machine Learning*, Vancouver, Canada. PMLR 267, 2025. Copyright 2025 by the author(s).

## 1. Introduction

Diffusion-based generative models (Sohl-Dickstein et al., 2015) have led to advances in modeling images (Ho et al., 2020; Song et al., 2020b), videos (Ho et al., 2022), and proteins (Gruver et al., 2023), as well as promising results for text generation (Li et al., 2022; Han et al., 2023; Gong et al., 2023; Gulrajani & Hashimoto, 2023; Horvitz et al., 2024). Despite these advances, diffusion models have failure modes. For example, text-to-image models often fail to adhere to text prompts (Ghosh et al., 2024). Additionally, adapting models to produce samples that conform to specific user preferences remains a challenge.

One approach for making generative models $p_\theta(\mathbf{x})$ adhere to user preferences is to encode preferences in a reward $r(\mathbf{x}_0)$ and sample from the *tilted* distribution $p_{target}(\mathbf{x}) \propto p_\theta(\mathbf{x}) \exp(r(\mathbf{x}))$ (Korbak et al., 2022), where $r(\mathbf{x})$ can be human preference models (Xu et al., 2024; Wu et al., 2023b), vision-language models (Liu et al., 2024a), or likelihoods $p(y \mid \mathbf{x})$ (Wu et al., 2023a). Sampling from this tilted distribution favors high-reward samples. Current approaches for sampling from the tilted distribution can be categorized into (a) fine-tuning and (b) inference-time steering methods.

Black et al. (2023), Fan et al. (2024), Domingo-Enrich et al. (2024), and Wallace et al. (2024) fine-tune diffusion models with reward functions. However, fine-tuning requires expensive training and ties the model to the reward used while training. Alternatively, two common inference-time approaches are gradient-based guidance (Song et al., 2020b; Bansal et al., 2023) and best-of-$n$ sampling. Best-of-$n$ sampling can be used for any diffusion model and reward function, however, it wastes computation on low-reward samples (Chatterjee & Diaconis, 2018). Gradient-based guidance presents an efficient alternative, but it is limited to differentiable reward functions and continuous-state diffusion models.

In this work, we present FK steering, a flexible framework for steering diffusion-based generative models with arbitrary rewards that uses FK interacting particle system methods (Moral, 2004; Vestal et al., 2008). We generalize previous works that define Feynman-Kac measures to condition-

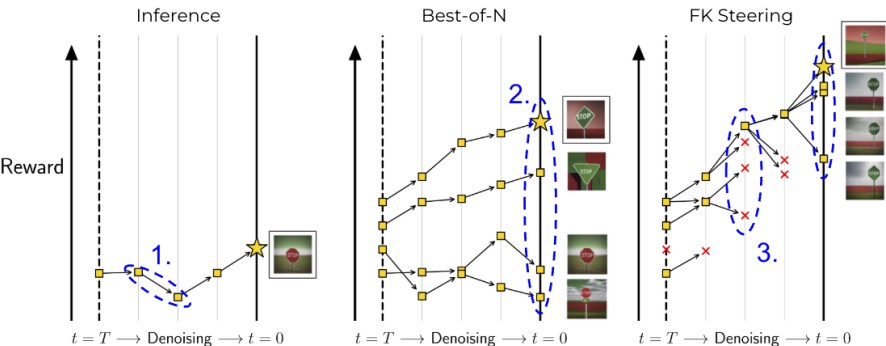

**Prompt**: *"a green stop sign in a red field"*

(1) Iteratively de-noise $x_T \to x_{T-1} \to ... \to x_0$.
(2) Generate multiple samples (*particles*).
(3) Resample promising particles at *intermediate* steps.

*Figure 1.* **Feynman-Kac steering** is a particle-based sampler which produces consistent approximations of the target distribution, $p_\theta(\mathbf{x}_0) \exp(\lambda r(\mathbf{x}_0))$. At intermediate steps, FK steering scores particles using functions called potentials and and then resamples based on potential scores. Potentials are defined using intermediate rewards and are selected such that paths yielding high-reward samples are up-weighted.

ally sample diffusion models (Trippe et al., 2022; Wu et al., 2023a; Chung et al., 2022; Janati et al., 2024). FK steering enables guidance with arbitrary reward functions, differentiable or otherwise, for both discrete and continuous-state models. The approach makes use of a rare-event simulation method, Feynman-Kac interacting particle system (FK-IPS) (Moral, 2004; Del Moral & Garnier, 2005; Hairer & Weare, 2014; Vestal et al., 2008). FK-IPS enables the generation of samples with high-rewards, which may be rare events under the original model $p_\theta(\mathbf{x})$.

Applying FK steering has two components: defining a sequence of tilted distributions over the diffusion trajectory using *potential* functions, and then sampling from these tilted distributions. To sample from these tilted distributions, FK steering (1) samples multiple diffusion processes, called *particles*, (2) scores particles using the potential functions, and (3) resamples the particles based on potential scores at intermediate steps during generation, see fig. 1. Potential functions are defined using intermediate rewards and are selected such that resampling high-scoring particles yield high-reward samples $\mathbf{x}_0$.

We show that diffusion models enable many choices of intermediate rewards, samplers, and potentials. We then empirically demonstrate that these new choices improve on traditional choices (Wu et al., 2023a). Remarkably, for a number of tasks, we see significant performance benefits for both image and text diffusion models with FK steering with as few as $k = 4$ particles (see fig. 2).

**Contributions.** Our methodological contributions are the following:

- We present Feynman-Kac steering, a flexible and effective framework for building particle-based approximations of the tilted distribution $p_\theta(\mathbf{x} \mid \mathbf{c}) \exp(\lambda r(\mathbf{x}, \mathbf{c}))$,

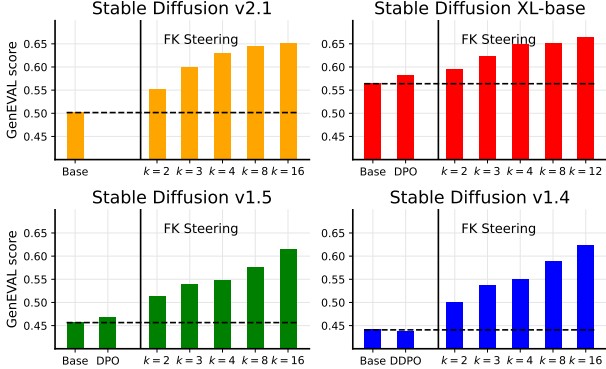

*Figure 2.* **FK steering small models outperforms bigger models with less compute**. We measure the prompt fidelity of samples from text-to-image models using the GenEval benchmark (Ghosh et al., 2024). We compare the highest-reward sample from FK steering the base models against the base models and their fine-tuned versions. As the reward, we use ImageReward (Xu et al., 2024). FK steering, with no extra training, improves performance for all models, outperforming fine-tuning with $k = 2$. Moreover, FK steering SDv2.1 (0.8B) outperforms a fine-tuned SDXL (2.6B) model, with fewer FLOPS and faster sampling.

for both continuous and discrete diffusion models, and for arbitrary rewards.

- We show that particle-based methods such as twisted diffusion sampler (TDS) (Wu et al., 2023a) and Li et al. (2024), are instances of FK interacting particle systems. Expanding the set of potentials, samplers, and reward models improves performance across many tasks.

Empirically, we demonstrate that FK steering:

- Provides an alternative to fine-tuning and gradient guidance. FK steering text-to-image diffusion models with

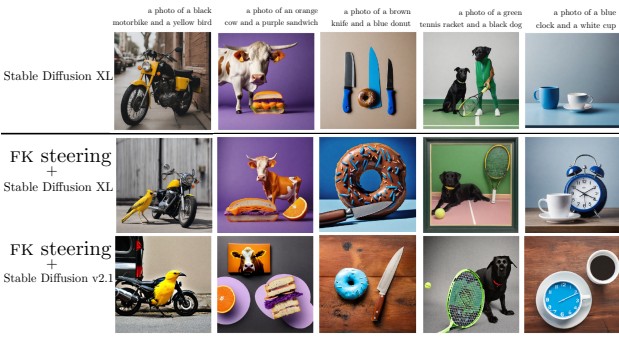

*Figure 3.* FK **steering improves prompt fidelity and sample quality.** *First row*: a random sample from SDXL. *Middle and bottom rows*: the highest reward sample using gradient-free FK steering with SDXL and SDv2.1, with $k = 4$. FK steering SDXL and SDv2.1 improves prompt fidelity compared to a random sample from the base model. Prompts are selected from the GenEval benchmark set.

human preference rewards outperforms fine-tuned models and gradient-guidance on a prompt fidelity benchmark with just two particles, see fig. 2. Moreover, FK steering combined with fine-tuned models or gradient guidance unlocks even further improvements. We also steer text diffusion models to generate higher quality samples with improved linguistic acceptability and perplexity.

- Enables smaller models to beat larger models (Ghosh et al., 2024), with faster sampling and less compute (see the right panel in fig. 2).

- Generates samples with (rare) specified attributes, such as toxicity, a useful attribute for red-teaming (Zhao et al., 2024a). FK steering a text diffusion model, without gradient guidance, increases the toxicity rate from $0.3\%$ to $64.7\%$, and outperforms best-of-$n$.

Overall, in *all* settings we consider, FK steering *always* improves performance, highlighting the benefits of inference-time scaling and steering of diffusion models.

## 2. Related Work

Current approaches to generate samples from the tilted distribution $p_\theta(\mathbf{x}_0)\exp(\lambda r(\mathbf{x}_0))$ can be categorized into two types: (1) fine-tuning and (2) inference-time steering approaches, such as universal guidance (Song et al., 2020b; Bansal et al., 2023) and particle-based approaches such as best-of-$n$ and TDS (Wu et al., 2023a).

**Fine-tuning.** Recent work (Black et al., 2023; Xu et al., 2024) proposes fine-tuning a diffusion model to maximize the reward without a Kullback-Leibler (KL) penalty. Fan et al. (2024); Domingo-Enrich et al. (2024) propose KL-regularized fine-tuning, and more recently Wallace et al.

(2024) propose direct preference optimization (Rafailov et al., 2024) for diffusion models. However, fine-tuning requires allocating training resources and coupling a model to a specific reward. Moreover, we show FK steering, with just 2 particles, outperforms fine-tuning in several settings without any additional training.

**Inference-time steering.** Gradient-based methods such as classifier guidance (Song et al., 2020b; Bansal et al., 2023) enable steering diffusion models at inference-time. Reward gradients are used to tilt the diffusion model's score, $s_\theta(\mathbf{x}_t, t) + \nabla_{\mathbf{x}_t} r(\mathbf{x}_t)$, where $s_\theta$ is the marginal score. However, gradient-based guidance is limited to differentiable rewards and continuous-state models.

FK steering builds on top of recent works that sample from Feynman-Kac path distributions for conditional sampling with diffusion models, either using particle-based sampling (Trippe et al., 2022; Wu et al., 2023a; Cardoso et al., 2023; Dou & Song, 2024; Zhao et al., 2024b) or gradient-based sampling (Chung et al., 2022; Janati et al., 2024). In appendix F.2, we show how TDS (Wu et al., 2023a) and SVDD (Li et al., 2024) are examples of FK interacting particle systems (Moral, 2004). Our experiments demonstrate the effectiveness of these methods for new settings, and the value of expanding the choice of potentials, rewards, and samplers.

## 3. Feynman-Kac Steering of Diffusion Models

In this section, we present details of the FK steering framework for inference-time steering of diffusion models.

### 3.1. Diffusion Models

Diffusion models (Sohl-Dickstein et al., 2015) are stochastic processes that are learned by reversing a forward noising process, $q(\mathbf{x}_t)$. The noising process takes data $x \sim q_{\text{data}}$ and produces a noisy state $\mathbf{x}_t \sim q(\mathbf{x}_t \mid \mathbf{x}_0 = x)$ such that at a terminal-time $T$, $q(\mathbf{x}_T) = \pi_{\text{prior}}$, where $\pi_{\text{prior}}$ is the model prior. The noising process can be defined as a continuous-time Markov process (Song et al., 2020b; Kingma et al., 2021; Singhal et al., 2023; 2024) or discrete-time Markov chain (Austin et al., 2021; Sahoo et al., 2024; Shi et al., 2024; Campbell et al., 2022). For exposition, we focus on discrete-time models, though the techniques are applicable to continuous-time as noted below. A discrete-time diffusion model, given a context $\mathbf{c}$, is defined as:

$$p_\theta(\mathbf{x}_T, \ldots, \mathbf{x}_0 \mid \mathbf{c}) = \pi_{\text{prior}}(\mathbf{x}_T) \prod_{t=T-1}^{0} p_\theta(\mathbf{x}_t \mid \mathbf{x}_{t+1}, \mathbf{c}).$$

Sampling involves iteratively generating a *path* $(\mathbf{x}_T, \mathbf{x}_{T-1}, \ldots, \mathbf{x}_0)$, where $\mathbf{x}_0$ is the model sample. The model $p_\theta$ can be trained by maximizing a lower bound on the model log-likelihood $\log p_\theta(\mathbf{x}_0 = x)$.

Most uses of generative models require samples with user-specified properties. In the next section, we describe a generic formulation for steering diffusion models towards such samples.

## 3.2. Steering Diffusion Models

One way to steer diffusion models is to encode user preferences in a reward model $r(\mathbf{x}_0)$ and sample from a distribution that tilts the diffusion model's generations $p_\theta(\mathbf{x}_0)$ towards an exponential of the reward function $r(\mathbf{x}_0)$:

$$p_{\text{target}}(\mathbf{x}_0 \mid \mathbf{c}) = \frac{1}{\mathbf{Z}} p_\theta(\mathbf{x}_0 \mid \mathbf{c}) \exp\left(\lambda r(\mathbf{x}_0, \mathbf{c})\right). \quad (1)$$

The reward can be any arbitrary function, such as a human preference reward (Xu et al., 2024; Wu et al., 2023b), a non-differentiable constraint, or a likelihood $p(y \mid \mathbf{x}_0)$.

The target distribution favors high-reward samples, which may be rare under the model $p_\theta$. This suggests the use of simulation methods that better tilt towards rare events. One broad class of rare-event simulation methods are FK-IPS approaches (Moral, 2004; Hairer & Weare, 2014) that *tilt the transition kernels* of the diffusion process to up-weight paths that have higher-reward samples.

Next, we develop FK steering, a framework for inference-time steering of diffusion models using FK-IPS.

## 3.3. Feynman-Kac diffusion steering

We use FK-IPS to produce paths $(\mathbf{x}_T, \mathbf{x}_{T-1}, \ldots, \mathbf{x}_0)$ with high-reward $\mathbf{x}_0$ samples. FK-IPS requires defining a sequence of FK distributions, $p_{\text{FK},t}(\mathbf{x}_T, \mathbf{x}_{T-1}, \ldots, \mathbf{x}_t)$, by tilting the base distribution $p_\theta(\mathbf{x}_T, \mathbf{x}_{T-1}, \ldots, \mathbf{x}_t)$ using potentials $G_t$ (Moral, 2004; Chopin et al., 2020). The sequence of distributions $p_{\text{FK},t}$ is built iteratively by tilting the transition kernels $p_\theta(\mathbf{x}_t \mid \mathbf{x}_{t+1})$ with a potential $G_t(\mathbf{x}_T, \mathbf{x}_{T-1}, \ldots, \mathbf{x}_t)$. We start with $p_{\text{FK},T}(\mathbf{x}_T) \propto p_\theta(\mathbf{x}_T \mid \mathbf{c}) G_T(\mathbf{x}_T, \mathbf{c})$ and then define the subsequent distributions as:

$$p_{\text{FK},t}(\mathbf{x}_T, \ldots, \mathbf{x}_t \mid \mathbf{c}) \quad (2)$$
$$= \frac{1}{\mathbf{Z}_t} p_\theta(\mathbf{x}_T, \ldots, \mathbf{x}_t \mid \mathbf{c}) \left\{ \prod_{s=T}^{t} G_t(\mathbf{x}_T, \ldots, \mathbf{x}_s, \mathbf{c}) \right\}$$

where $\mathbf{Z}_t = \mathbb{E}_{p_\theta}[\prod_{s=T}^{t} G_s]$ is the normalization constant. The potentials $G_t$ are selected to up-weight paths $(\mathbf{x}_T, \ldots, \mathbf{x}_t)$ that yield high-rewards $r(\mathbf{x}_0)$. We require that the product of the potentials $G_t$ matches the exponential tilt of $p_{\text{target}}$:

$$\prod_{t=T}^{0} G_t(\mathbf{x}_T, \ldots, \mathbf{x}_t, \mathbf{c}) = \exp\left(\lambda r(\mathbf{x}_0, \mathbf{c})\right). \quad (3)$$

This choice ensures that sampling $\mathbf{x}_0$ from $p_{\text{FK},0}$ is equivalent to sampling $p_{\text{target}}(\mathbf{x}_0 \mid \mathbf{c})$, since $p_{\text{FK},0} \propto$

---

**Algorithm 1** Feynman-Kac Diffusion Steering

**Input:** Diffusion model $p_\theta(\mathbf{x}_{0:T} \mid \mathbf{c})$, reward $r(\mathbf{x}_0, \mathbf{c})$, proposals $\tau(\mathbf{x}_t \mid \mathbf{x}_{t+1}, \mathbf{c})$, potentials $G_t$, intermediate rewards $r_\phi(\mathbf{x}_t, \mathbf{c})$, number of particles $k$.
**Sample** $\mathbf{x}_T^i \sim \tau(\mathbf{x}_T \mid \mathbf{c})$ for $i \in [K]$
**Score** $G_T^i = G_T(\mathbf{x}_T^i, \mathbf{c})$ for $i \in [K]$
**for** $t \in \{T, \ldots, 1\}$ **do**
  **Resample:** Sample $a_t^i \sim \text{Multinomial}(\mathbf{x}_t^i, G_t^i)$ and let $\mathbf{x}_t^i = \mathbf{x}_t^{a_i}$ for $i \in [K]$
  **Propose:** Sample $\mathbf{x}_{t-1}^i \sim \tau(\mathbf{x}_{t-1} \mid \mathbf{x}_t^i, \ldots, \mathbf{x}_T^i, \mathbf{c})$ for $i \in [K]$
  **Re-weight:** Compute weight $G_{t-1}^i$ for $i \in [K]$:
$$G_{t-1}^i = \frac{p_\theta(\mathbf{x}_{t-1}^i \mid \mathbf{x}_t^i, \mathbf{c})}{\tau(\mathbf{x}_{t-1}^i \mid \mathbf{x}_{t:T}^i, \mathbf{c})} G_{t-1}(\mathbf{x}_{T:t-1}^i, \mathbf{c})$$
**end for**
**Output:** return samples $\{\mathbf{x}_0^i\}$

---

$p_\theta(\mathbf{x}_T, \ldots, \mathbf{x}_0 \mid \mathbf{c}) \exp(\lambda r(\mathbf{x}_0, \mathbf{c}))$. Potential functions that satisfy this constraint are not unique.

**Sampling from $p_{\text{FK},0}$.** Direct sampling from the FK measure, $p_{\text{FK},0}$, is intractable. However, targeting the intermediate distributions $p_{\text{FK},t}$ supports sampling of the distribution $p_{\text{FK},0}$ with particle-based methods, such as sequential Monte Carlo (SMC) (Moral, 2004; Doucet & Lee, 2018), nested IS (Naesseth et al., 2019), and diffusion Monte Carlo (DMC) (Hairer & Weare, 2014). SMC generates $k$ particles using a proposal generator $\tau(\mathbf{x}_t \mid \mathbf{x}_{t+1}, \ldots, \mathbf{x}_T, \mathbf{c})$ and at each transition step scores the particles using the potential and the transition kernel importance weights:

$$G_t^i = G_t(\mathbf{x}_T^i, \ldots, \mathbf{x}_{t+1}^i, \mathbf{x}_t^i, \mathbf{c}) \frac{p_\theta(\mathbf{x}_t^i \mid \mathbf{x}_{t+1}^i, \mathbf{c})}{\tau(\mathbf{x}_t^i \mid \mathbf{x}_{t+1}^i, \ldots, \mathbf{x}_T^i, \mathbf{c})}.$$

Next, the particles $\mathbf{x}_t^i$ are resampled based on the scores $G_t^i$. See algorithm 1 for details. Particle approximations are consistent, that is the weighted empirical distribution defined by $((\mathbf{x}_T^i, \ldots, \mathbf{x}_t^i), G_t^i)$ converges to $p_{\text{FK},t}$, see theorem 3.19 in Del Moral & Miclo (2000). For a proof that the weighted empirical distribution, $(\mathbf{x}_0^i, G_t^i)$, converges to $p_{\text{target}}$, see appendix C.

**Choosing the proposal generator $\tau$.** For the proposal generator $\tau$, the simplest choice is to sample from the diffusion model's transition kernel $p_\theta(\mathbf{x}_t \mid \mathbf{x}_{t+1}, \mathbf{c})$. Alternatively, another choice is to tilt the transition kernels towards high-reward samples, for instance, by using reward-gradient guidance (Song et al., 2020b; Bansal et al., 2023). We discuss some choices in appendix D.1.

**Choosing the potential $G_t$.** One choice of potentials is $G_t = 1$ for $t \geq 1$ and $G_0 = \exp(\lambda(r(\mathbf{x}_0, \mathbf{c})))$, this leads to importance sampling. However, importance sampling

can require many particles to generate a high-reward sample (Chatterjee & Diaconis, 2018). Instead, FK steering uses potentials to up-weight paths that yield high-reward samples. We consider the following potentials that satisfy eq. (3), defined using *intermediate rewards* $r_\phi(\mathbf{x}_t, \mathbf{c})$:

- DIFFERENCE: $G_t(\mathbf{x}_t, \mathbf{x}_{t+1}, \mathbf{c}) = \exp(\lambda(r_\phi(\mathbf{x}_t, \mathbf{c}) - r_\phi(\mathbf{x}_{t+1}, \mathbf{c})))$ and $G_T = 1$, similar to (Wu et al., 2023a), prefers particles that have increasing rewards.

- MAX: $G_t(\mathbf{x}_T, \ldots, \mathbf{x}_t, \mathbf{c}) = \exp(\lambda \max_{s=t}^T r_\phi(\mathbf{x}_s, \mathbf{c}))$ and $G_0 = \exp(\lambda r(\mathbf{x}_0, \mathbf{c}))(\prod_{t=1}^T G_t)^{-1}$ prefers particles that have the highest rewards.

- SUM: $G_t(\mathbf{x}_T, \ldots, \mathbf{x}_t) = \exp(\lambda \sum_{s=t}^T r_\phi(\mathbf{x}_s, \mathbf{c}))$ and $G_0 = \exp(\lambda r(\mathbf{x}_0, \mathbf{c}))(\prod_{t=1}^T G_t)^{-1}$ selects particles that have the highest accumulated rewards.

Any choice of potentials that satisfy eq. (3) produce consistent approximations of $p_{\text{target}}(\mathbf{x}_0)$. However, the rewards of the particle approximation depend on the choice of potentials. For instance, if $r(\mathbf{x}_0)$ is bounded, then using the difference potential assigns low scores to particles that reach the maximum reward early in generation. In this setting, alternatives like the MAX potential may be apt.

*Interval Resampling.* For a typical diffusion process, the states $\mathbf{x}_t$ and $\mathbf{x}_{t+1}$ do not differ significantly. As a result, we propose interval resampling. We resample at *selected* steps, specified by a *resampling* schedule $R = \{t_r, \ldots, 0\}$. For $t \in R$, $G_t$ is a non-uniform potential, such as the max potential, otherwise $G_t = 1$. Interval resampling encourages exploration and reduces sampling time and compute. See fig. 8 and appendix E for its effect on samples.

**Choosing intermediate rewards** $r_\phi(\mathbf{x}_t, \mathbf{c})$**.** The ideal rewards for the intermediate state $\mathbf{x}_t$ requires knowledge of the distribution of terminal rewards given an intermediate step, $p_\theta(r(\mathbf{x}_0) \mid \mathbf{x}_t, \mathbf{c})$. With this distribution, rewards $r_\phi$ can be chosen to ensure high-expected rewards or good worst-case quality by using the 10th percentile. Producing this distribution of rewards requires training with model samples, which can be expensive. Alternatively, we demonstrate that diffusion models offer many options with different trade-offs between compute versus the quality of the reward estimate $r(\mathbf{x}_0)$:

- **Rewards at expected** $\mathbf{x}_0$**.** Similar to Song et al. (2020b); Bansal et al. (2023); Wu et al. (2023a); Li et al. (2024), intermediate rewards can be defined by evaluating the reward function at the diffusion model's approximation of the expected sample $\mathbf{x}_0$: $\widehat{\mathbf{x}}_t \approx \mathbb{E}_{p_\theta(\mathbf{x}_0 \mid \mathbf{x}_t, \mathbf{c})}[\mathbf{x}_0 \mid \mathbf{x}_t, \mathbf{c}]$. With this choice, the intermediate rewards are $r_\phi(\mathbf{x}_t, \mathbf{c}) = r(\mathbf{x}_0 = \widehat{\mathbf{x}}_t, \mathbf{c})$.

- **Many-sample** $r_\phi$**.** Diffusion models provide a means to sample $p_\theta(\mathbf{x}_0 \mid \mathbf{x}_t, \mathbf{c})$. During inference, for each particle

$\mathbf{x}_t^i$, we sample $N$ samples $\mathbf{x}_0^{i,j} \sim p_\theta(\mathbf{x}_0 \mid \mathbf{x}_t^i, \mathbf{c})$ and then use $r_\phi(\mathbf{x}_t^i, \mathbf{c}) = \log \frac{1}{N} \sum_{j=1}^N \exp(r(\mathbf{x}_0^{i,j}, \mathbf{c}))$ to summarize the empirical distribution of rewards.

- **Learned** $r_\phi$**.** When sampling from $p_\theta(\mathbf{x}_0 \mid \mathbf{x}_t, \mathbf{c})$ is expensive, we can use the fact that $p_\theta$ is trained to approximate the noise process $q$ (Sohl-Dickstein et al., 2015; Song et al., 2020b). Therefore, we can use data samples to train $r_\phi$. For instance, when $r(\mathbf{x}_0)$ is a classifier $p_\theta(y \mid \mathbf{x}_0)$, then Nichol et al. (2021) train a classifier $p_\phi(y \mid \mathbf{x}_t)$. For more general rewards, we can use:

$$\mathbb{E}_{t \sim U[0,T]} \mathbb{E}_{q_{\text{data}}(\mathbf{x}_0) q(\mathbf{x}_t \mid \mathbf{x}_0)} \|a_\phi(\mathbf{x}_t, \mathbf{c}) - \exp(r(\mathbf{x}_0, \mathbf{c}))\|_2^2$$

and define $r_\phi = \log a_\phi$. When $p_\theta = q$, the reward $r_\phi = \log \mathbb{E}_{p_\theta(\mathbf{x}_0 \mid \mathbf{x}_t, \mathbf{c})}[\exp(r(\mathbf{x}_0, \mathbf{c}))]$ can be used to define potentials $G_t$ that leads to the local transitions which minimize the variance of the potential at each step, see theorem 10.1 in (Chopin et al., 2020).

We note that as long as the potentials satisfy eq. (3), *any* choice of $r_\phi$ allows for consistent approximations. See fig. 5 for how different choices of $r_\phi$ correlate with $r(\mathbf{x}_0)$.

**Continuous-time diffusions.** While the presentation above is for discrete-time models, FK steering can also be used for continuous-time models (Song et al., 2020b; Kingma et al., 2021; Singhal et al., 2023). Continuous-time models are sampled using numerical methods, such as Euler-Maruyama (Särkkä & Solin, 2019), which involve defining a discrete grid $\{1, 1 - \Delta t, \ldots, 0\}$ and then sampling from the transition kernel $p_\theta(\mathbf{x}_t \mid \mathbf{x}_{t+\Delta}, \mathbf{c})$. Therefore, similar to discrete-time models, FK steering can tilt the transition kernels with potentials $G_t(\mathbf{x}_1, \mathbf{x}_{1-\Delta t}, \ldots, \mathbf{x}_t)$.

## 4. Experiments

We evaluate FK steering with the following experiments:

- **FK steering for sample quality**: This experiment steers text-to-image diffusion models and text diffusion models with rewards that measure sample quality.

  - For text-to-image models, we use a human preference score, ImageReward, as the reward function. We evaluate on the heldout GenEval benchmark, a prompt fidelity benchmark.
  - For text diffusion models, we explore three choices of rewards: the perplexity computed using either GPT2 (Radford et al., 2019) or a trigram language model (Liu et al., 2024b), and a linguistic acceptability classifier (Morris et al., 2020).

- **Studying potential choices in FK steering**: Here we study the effect of the choices of potential on the rewards $r(\mathbf{x}_0^i)$.

- **Studying different choices of intermediate rewards**: We examine the effect of using different intermediate rewards with FK steering.

    - For text diffusion models, we consider control of text *toxicity*, which occurs in around $1\%$ of base model samples.

    - For image diffusion models, we do class-conditional generation on ImageNet. In this experiment, we incorporate reward gradients to tilt the proposal generator.

### 4.1. FK steering for sample quality

**Text-to-Image Diffusion Models.** Here we use stable diffusion (Rombach et al., 2022; Podell et al., 2023; von Platen et al., 2022) text-to-image models $p_\theta(\mathbf{x}_0 \mid \mathbf{c})$, where $\mathbf{c}$ is the text prompt. These models include both continuous and discrete-time processes. As the reward, we use the ImageReward preference model (Xu et al., 2024). Intermediate rewards are defined by evaluating the reward model on the denoised state, $r_\phi(\mathbf{x}_t) = r(\mathbf{x}_0 = \widehat{\mathbf{x}}_t)$ where $\widehat{\mathbf{x}}_t \approx \mathbb{E}_{p_\theta}[\mathbf{x}_0 \mid \mathbf{x}_t]$.

For the proposal generator $\tau$, we use the base model itself. For sampling from the base model, we use classifier-free guidance (Ho & Salimans, 2022) with guidance scale set to $7.5$[1], alongside the DDIM sampler (Song et al., 2020a) with $\eta = 1$ and $T = 100$ time-steps. We use $\lambda = 10$ and resampling schedule $[0, 20, 40, 60, 80]$ with the max potential $\exp(\lambda \max_{s=t}^{T} r_\phi(\mathbf{x}_s))$, see table 7 for score model parameter counts and sampling time.

We measure prompt fidelity using the GenEval benchmark[2] (Ghosh et al., 2024) and we also report ImageReward[3] and HPS (Wu et al., 2023b) scores. See appendix A for results with different sampling choices. As a benchmark, we compare against best-of-$n$ (BoN) sampling and gradient guidance. Additionally, we also benchmark against publicly available models, fine-tuned for prompt alignment and aesthetic quality. We use models fine-tuned using DPO[4] (Wallace et al., 2024) and DDPO (Black et al., 2023)[5], an RL-based method. Additionally, we also evaluate FK steering fine-tuned models.

In table 1 we report the prompt fidelity and aesthetic quality scores of the highest-reward particle generated by FK steering, and in fig. 4, we report average particle performance.

---

[1]Default choice from Hugging Face, see https://huggingface.co/blog/stable_diffusion

[2]Prompts from https://github.com/djghosh13/geneval/tree/main/prompts

[3]Prompts from https://github.com/THUDM/ImageReward/blob/main/data/test.json

[4]https://huggingface.co/papers/2311.12908

[5]https://huggingface.co/kvablack/ddpo-alignment

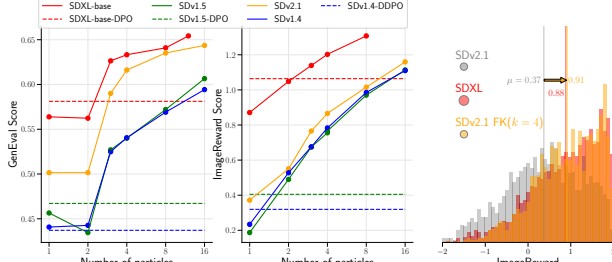

*Figure 4.* **Effect of scaling the number of particles.** *Left*: GENEVAL scores for FK steering using IMAGEREWARD, average particle performance. Dashed lines indicate performance of fine-tuned baselines. *Middle*: Corresponding IMAGEREWARD scores. *Right*: Distribution of IMAGEREWARD scores for samples from SDv2.1 (0.8B) with and without FK steering, compared with SDXL (2.6B).

We observe:

- **FK steering the base model beats fine-tuning.** FK steering with $k = 4$ particles outperforms fine-tuned models on both prompt fidelity and human preference alignment. Moreover, Figure 2 shows that FK steering with just $k = 2$ has a *higher* GenEval score than the DPO and DDPO fine-tuned models. Additionally, we show that in Table 4, FK steering outperforms steering with gradient guidance (Bansal et al., 2023) using the ImageReward model.

- **FK steering smaller models outperforms larger models.** With $k = 4$, FK steering SDv2.1 outperforms SDXL and its DPO (Wallace et al., 2024) fine-tuned version, on GenEval scores and aesthetic quality with less sampling time: 11.5s versus 9.1s, see fig. 3 for samples.

- **Steering fine-tuned models.** In table 1, we observe that FK steering fine-tuned models further improves performance.

- **Effect of scaling the number of particles.** Figure 4 shows that scaling the number of particles improves the average prompt fidelity and human preference alignment scores of all particles for all models.

**Text Diffusion Models.** Next, we investigate steering to improve the sample quality of text diffusion models (Li et al., 2022; Gulrajani & Hashimoto, 2023; Horvitz et al., 2024). We consider two base text diffusion models: SSD-LM (Han et al., 2023) and MDLM (Sahoo et al., 2024) and use these models as the proposal generator $\tau$. SSD-LM is a continuous space diffusion model trained on noised word logits, while MDLM is a discrete diffusion model. We consider three reward functions for improving text quality: perplexity computed with a *trigram lan-*

| Model | Sampler | GenEval[2] ↑ | IR[3] ↑ | HPS[3] ↑ |
|---|---|---|---|---|
| SDv1.4 | $k = 1$ | 0.44 | 0.23 | 0.245 |
| SDv1.4 | BoN($k = 4$) | 0.54 | 0.80 | 0.256 |
| SDv1.4$_{\text{DDPO}}$ | $k = 1$ | 0.43 | 0.26 | 0.241 |
| **SDv1.4** | FK($k = 4$) | **0.54** | **0.92** | **0.26** |
| SDv1.5 | $k = 1$ | 0.44 | 0.18 | 0.245 |
| SDv1.5 | $\nabla(k = 1)$ | 0.45 | 0.66 | 0.245 |
| SDv1.5 | BoN($k = 4$) | 0.52 | 0.73 | 0.265 |
| SDv1.5$_{\text{DPO}}$ | $k = 1$ | 0.46 | 0.34 | 0.255 |
| **SDv1.5** | FK($k = 4$) | **0.54** | **0.89** | **0.263** |
| **SDv1.5$_{\text{DPO}}$** | FK($k = 4$) | **0.57** | 0.88 | **0.276** |
| SDv2.1 | $k = 1$ | 0.51 | 0.37 | 0.253 |
| SDv2.1 | BoN($k = 4$) | 0.61 | 0.88 | 0.263 |
| **SDv2.1** | FK($k = 3$) | 0.59 | 0.86 | 0.265 |
| **SDv2.1** | FK($k = 4$) | **0.62** | **1.01** | **0.268** |
| SDXL | $k = 1$ | 0.55 | 0.87 | 0.289 |
| SDXL | BoN($k = 4$) | 0.63 | 1.23 | 0.296 |
| SDXL$_{\text{DPO}}$ | $k = 1$ | 0.58 | 0.85 | 0.296 |
| **SDXL** | FK($k = 4$) | **0.64** | **1.29** | **0.302** |
| **SDXL$_{\text{DPO}}$** | FK($k = 4$) | **0.67** | **1.19** | **0.317** |

*Table 1.* **Effect of FK steering on prompt fidelity and human preference scores:** For all models, FK steering improves performance, outperforming best-of-$n$, gradient guidance ($\nabla$), and fine-tuning. Interestingly, even best-of-$n$ outperforms fine-tuning, showing the effectiveness of inference-time scaling. For all metrics, a higher value is better.

*guage model*[6], a classifier[7] (Morris et al., 2020) trained on the Corpus of Linguistic Acceptability (CoLA) dataset (Warstadt et al., 2018), and perplexity computed by GPT2. For all choices of reward models, we define the intermediate rewards using $r_\phi(\mathbf{x}_t) = r(\mathbf{x}_0 = \hat{\mathbf{x}}_t)$ and the potential $G_t = \exp(\lambda(r_\phi(\mathbf{x}_t) - r_\phi(\mathbf{x}_{t+1})))$.

For both models, we resample 50 times, every 10 steps for SSD-LM ($T = 500$) and every 20 for MDLM ($T = 1000$). We use $\lambda = 10.0$ and return the highest reward sample at $t = 0$. Following Han et al. (2023), we generate 20 continuations of length 50 using their 15 prompts. In addition, we evaluate base model performance, best-of-$n$, and GPT2-Medium performance. As a baseline, we also include results for SSD-LM with more sampling time-steps, $T = 5000$ versus $T = 500$ for FK steering. We evaluate perplexity using GPT2-XL and CoLA acceptability. Additional details are included in appendix B.

Table 2 contains the evaluation results. We observe:

- **FK steering improves the perplexity and CoLA scores of both models**. For all reward functions, FK steering with $k = 4$ outperforms best-of-4 on the corresponding target metric (perplexity or CoLA). For MDLM, trigram

---

[6] We compute trigram probabilities using $\infty$-gram (Liu et al., 2024b).

[7] https://huggingface.co/textattack/roberta-base-CoLA

---

| Model + Sampler($r$) | $k$ | PPL (GPT-XL) ↓ | CoLA ↑ |
|---|---|---|---|
| GPT2-medium | 1 | 14.1 | 87.6 |
| SSD-LM | 1 | 23.2 | 68.3 |
| SSD-LM$_{T \times 10}$ | 1 | 18.8 | 76.6 |
| **FK(GPT2)** | 4 | **11.0** | 80.0 |
| **FK(Trigram)** | 4 | 14.1 | 77.4 |
| **FK(CoLA)** | 4 | 17.4 | 95.7 |
| BoN(GPT2) | 4 | 13.6 | 75.6 |
| BoN(Trigram) | 4 | 15.9 | 71.9 |
| BoN(CoLA) | 4 | 19.2 | 93.8 |
| BoN(GPT2) | 8 | 11.2 | 80.3 |
| BoN(Trigram) | 8 | 13.9 | 76.8 |
| BoN(CoLA) | 8 | 18.4 | **97.2** |
| MDLM | 1 | 85.3 | 28.9 |
| **FK(GPT2)** | 4 | 49.0 | 39.8 |
| **FK(Trigram)** | 4 | **40.3** | 37.0 |
| **FK(CoLA)** | 4 | 73.6 | 69.8 |
| BoN(GPT2) | 4 | 55.5 | 32.9 |
| BoN(Trigram) | 4 | 52.1 | 30.1 |
| BoN(CoLA) | 4 | 71.4 | 59.4 |
| BoN(GPT2) | 8 | 46.9 | 37.2 |
| BoN(Trigram) | 8 | 45.9 | 35.4 |
| BoN(CoLA) | 8 | 68.2 | **73.1** |

*Table 2.* **Text sample quality results metrics.** We sample texts of length 50 from all models and score perplexity with GPT2-XL and CoLA acceptability. Results are averaged over three seeds. Both SSD-LM and GPT-medium have 355 million parameters. MDLM is a smaller model with 170 million parameters.

steering dramatically improves perplexity (40.3 vs 85.3), but is less effective at improving CoLA (37.0 vs 28.9).

- **FK steering outperforms best-of-$n$.** For all settings, FK steering outperforms best-of-$n$ for the same number of particles. Notably, in many cases FK steering outperforms best-of-$n$ with twice as many particles. Additionally, FK steering SSD-LM with $T = 500$ outperforms SSD-LM with $T = 5000$ for all metrics.

Overall, our results demonstrate that FK steering with off-the-shelf rewards can enable sampling lower-perplexity, more linguistically acceptable text from diffusion models.

### 4.2. Studying different choices of potentials

In the previous section, we use two different potentials: the max potential, $\exp(\lambda \max_{s \geq t} r_\phi(\mathbf{x}_s))$, for the text-to-image experiments and the difference potential, $\exp(\lambda(r_\phi(\mathbf{x}_t) - r_\phi(\mathbf{x}_{t+1})))$, for the text quality experiment. However, as discussed in section 3, the choice of potential is not unique. In this experiment, we steer text-to-image diffusion models with different choices of potentials, including the sum, max and difference potentials.

In table 3, for all models, using the max potential yielded higher prompt fidelity scores. Since ImageReward is

| Potential | $k$ | SDv1.4 | SDv1.5 | SDv2.1 | SDXL |
|---|---|---|---|---|---|
| Max | 4 | **0.540** | **0.540** | **0.616** | **0.633** |
| Sum | 4 | 0.496 | 0.499 | 0.569 | 0.613 |
| Difference | 4 | 0.525 | 0.526 | 0.578 | 0.603 |
| Max | 8 | **0.569** | **0.561** | **0.635** | **0.648** |
| Sum | 8 | 0.532 | 0.517 | 0.588 | 0.634 |
| Difference | 8 | 0.566 | 0.553 | 0.615 | 0.640 |

*Table 3.* **Effect of different potentials on GenEval scores.** Here we have the GenEval prompt fidelity score, averaged over all particles. Using the max potential outperforms the difference potential and the sum potential.

| Model | Sampler | GenEval | IR | HPS | Time |
|---|---|---|---|---|---|
| SDv1.5 | $k = 1$ | 0.44 | 0.187 | 0.245 | 2.4s |
| SDv1.5 | $\nabla(k = 1)$ | 0.45 | 0.668 | 0.245 | 20s |
| SDv1.5 | FK$(k = 4)$ | 0.54 | 0.898 | 0.263 | 8.1s |
| SDv1.5 | FK$(\nabla, k = 4)$ | 0.56 | 1.290 | 0.268 | 55s |

*Table 4.* **Comparison against gradient guidance.** Here we note that FK steering with the model as the proposal generator outperforms gradient guidance, with faster sampling. We also note that FK steering can benefit from gradient guidance, albeit at the cost of more compute and sampling time.

bounded between $[-2, 2]$, using the difference of intermediate rewards can assign lower scores to particles that achieve the maximum reward early in generation. However, for the same $\lambda$ and $k$, the max potential favors higher scoring particles more so than the difference potential. This can lead to lower particle diversity. See appendix E for samples.

### 4.3. Studying different choices of intermediate rewards

In this experiment, we study the effect of using different choices of intermediate rewards on FK steering. Here we generate samples with rare attributes, such as (a) toxicity for text diffusion models and (b) class-conditional image generation with 1000 classes in the dataset.

**Controlling Text Toxicity.** We consider the task of red-teaming *toxicity*, a rare attribute identified in only $1\%$ of base SSD-LM samples and $0.3\%$ of MDLM samples. Here, we examine whether FK steering enables testing rare but dangerous model behavior, a critical factor considered before deploying systems (Zhao et al., 2024a). The text diffusion models, SSD-LM and MDLM, the sampling parameters, and prompts are identical to section 4.1. We use the base models as the proposal generators. As a baseline, we compare against gradient guidance for SSD-LM and best-of-$n$ for both models. For reward, we use a popular toxicity

| Model + Sampler | Toxic ↑ | Toxic (H) ↑ | PPL ↓ |
|---|---|---|---|
| SSD-LM | 0.4% | 1.2% | **23.2** |
| SSD-LM ($\nabla$ guidance) | 22.3% | 22.6% | 40.3 |
| MDLM | 0.3% | 1.9% | 85.3 |
| SSD-LM (no gradients) | | | |
| BoN(4) | 1.6% | 4.8% | 21.9 |
| BoN(8) | 5.0% | 8.1% | 23.0 |
| **FK**$(k = 4)$ | 8.4% | 14.0% | 22.5 |
| **FK**$(k = 4$, **learned** $r_\phi)$ | 15.2% | 19.6% | 26.3 |
| **FK**$(k = 8)$ | 25.0% | 29.7% | 23.9 |
| **FK**$(k = 8$, **learned** $r_\phi)$ | **39.0%** | **38.0%** | 26.9 |
| MDLM (no gradients) | | | |
| BoN(4) | 2.2% | 6.7% | 83.8 |
| BoN(8) | 3.7% | 10.8% | 84.6 |
| **FK**$(k = 4)$ | 23.0% | 29.0% | 81.0 |
| **FK**$(k = 4$, **many** $r_\phi)$ | 37.0% | 40.2% | 83.0 |
| **FK**$(k = 8)$ | 53.4% | 48.3% | 74.3 |
| **FK**$(k = 8$, **many** $r_\phi)$ | **64.7%** | **51.7%** | 82.9 |

*Table 5.* **Toxicity results.** We evaluate the toxicity of the generated samples with (a) the classifier used for steering and (b) a separate holdout (H) classifier, we also report GPT2-XL perplexity.

classifier (Logacheva et al., 2022).[8]

In this experiment, we explore the effect of different choices of intermediate rewards:

- For SSD-LM, we consider two choices: (1) the reward evaluated at the denoised state and (2) the reward $r_\phi$ learned with real data.

- For MDLM, we use $N$ samples $\mathbf{x}_0^{i,j} \sim p_\theta(\mathbf{x}_0^{i,j} \mid \mathbf{x}_t^i)$ to compute the reward $r_\phi = \log \frac{1}{N} \sum_{j=1}^{N} \exp(r(\mathbf{x}_0^{i,j}))$ with $N = 4, 16$ samples.

For evaluation, we also include results from an *additional holdout* toxicity classifier, trained on a multilingual mixture of toxicity datasets (Dementieva et al., 2024).[9] Details are included in appendix B. In Table 5, we observe the following:

- **Using many-sample $r_\phi$ improves controllability:** FK steering MDLM with $k = 8$ achieves an accuracy of $53.4\%$. Using more samples for intermediate rewards improves performance even further to $64.7\%$. FK steering outperforms best-of-$n$ sampling with both 4 and 8 particles.

- **FK steering can outperform gradient guidance and preserves fluency:** With 8 particles, FK steering SSD-LM outperforms gradient guidance on holdout toxicity accuracy ($29.7\%$ vs $22.6\%$), and improves on perplexity

---

[8] https://huggingface.co/s-nlp/roberta_toxicity_classifier

[9] https://huggingface.co/textdetox/xlmr-large-toxicity-classifier

(23.9 vs 40.3). Using **learned** intermediate rewards improves performance further, increasing toxicity to 39.0%.

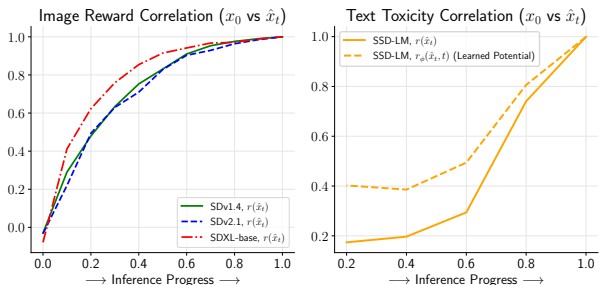

*Figure 5.* **Correlation between** $r_\phi(\mathbf{x}_t)$ **and final state** $r(\mathbf{x}_0)$: *Left:* Correlations between $r(\mathbf{x}_0)$ and $r(\mathbf{x}_0 = \widehat{\mathbf{x}}_t)$ for several text-to-image models, where $r$ is the ImageReward model. *Right*: Correlation between a text toxicity classifier $r(\mathbf{x}_0)$ and (a) $r(\mathbf{x}_0 = \widehat{\mathbf{x}}_t)$ and (b) learned $r_\phi(\mathbf{x}_t)$, using on SSD-LM. Learning the intermediate rewards with a regression objective improves the correlation between $r(\mathbf{x}_t)$ and $r(\mathbf{x}_0)$.

**Better Rewards vs. More Particles.** We observed that using better intermediate rewards, either learned or using multiple samples, improves performance. For instance, FK steering SSD-LM for $k = 4$ achieves 15.2% accuracy with learned rewards, compared to 8.4% when using the reward evaluated at the denoised state, however, with $k = 8$ accuracy increases to 25%, without the learned rewards. Therefore, FK steering offers two ways for scaling compute to improve performance: allocating additional resources to better estimate rewards $r(\mathbf{x}_0)$, or by scaling the number of particles.

**Class-Conditional Image Generation.** In this experiment, we steer a marginal diffusion model $p_\theta(\mathbf{x}_0)$ to produce samples from one of 1000 different classes. Similar to Wu et al. (2023a), the reward is $r(\mathbf{x}_0, y) = \log p_\theta(y \mid \mathbf{x}_0)$ and we also use *gradient guidance* for the proposal distribution $\tau(\mathbf{x}_t \mid \mathbf{x}_{t+1}, \mathbf{c})$.

We compare two potentials, the max potential and the difference potentials, along with two different reward models: one that uses the denoised state $r(\mathbf{x}_0 = \widehat{\mathbf{x}}_t, y)$ and one that is trained on noisy states $\mathbf{x}_t \sim q(\mathbf{x}_t \mid \mathbf{x}_0)$ where $\mathbf{x}_0 \sim q_{\text{data}}$ (Nichol et al., 2021). This experiment uses pre-trained marginal diffusion model and classifiers from Nichol & Dhariwal (2021) and generates $256 \times 256$ resolution images. In table 6, we observe that learning $r_\phi$, for both gradient guidance and potential computation, provides significant improvements over the reward evaluated at the denoised state.

## 5. Conclusion

We present Feynman-Kac steering, a framework for inference-time steering of diffusion modeling, based on

| $r_\phi(\mathbf{x}_t)$ | $G_t$ | $p(y \mid \mathbf{x}_0)$ **Mean (Max)** |
|---|---|---|
| $r(\mathbf{x}_0 = \widehat{\mathbf{x}}_t)$ | Diff. | 0.59 (0.72) |
| $r(\mathbf{x}_0 = \widehat{\mathbf{x}}_t)$ | Max | 0.65 (0.70) |
| Learned | Diff. | 0.88 (0.94) |
| Learned | Max | 0.88 (0.96) |

*Table 6.* **ImageNet class-conditional probabilities with different choices of rewards and potentials.** In this experiment, we explore the effect of two choices of rewards, learned and the reward evaluated at the denoised state (Wu et al., 2023a). We also explore the effect of different choices of potentials, the difference and the max potential. We observe that learning the reward improves performance significantly.

FK-IPS (Moral, 2004). Our experiments demonstrate that FK steering can improve sample quality and controllability of image and text diffusion models, outperforming fine-tuning and other inference-time approaches.

FK steering can be used in a "plug-and-play" fashion, with no extra training. For instance, using the difference potential with intermediate rewards defined using the denoised state and the base model as the proposal generator improves performance significantly, outperforms fine-tuned models, and enables small models to outperform larger models, *with less compute*. Additionally, by exploring different choices of potentials, intermediate rewards, and samplers, users can optimize performance for their tasks.

Our experiments show that scaling the number of particles is a natural mechanism for improving diffusion models. Notably, in our text-to-image experiments, *even best-of-4 outperforms fine-tuned models*. FK steering improves on best-of-$n$ by resampling using intermediate rewards during generation, resulting in efficient inference-time scaling.

## Acknowledgments

The authors would like to acknowledge Stefan Andreas Baumann, Yunfan Zhang, Anshuk Uppal, Mark Goldstein, and Eric Horvitz for their valuable feedback.

This work was partly supported by the NIH/NHLBI Award R01HL148248, NSF Award 1922658 NRT-HDR: FUTURE Foundations, Translation, and Responsibility for Data Science, NSF CAREER Award 2145542, ONR N00014-23-1-2634, and Apple. Additional support was provided by a Fellowship from the Columbia Center of AI Technology. This work was also supported by IITP with a grant funded by the MSIT of the Republic of Korea in connection with the Global AI Frontier Lab International Collaborative Research (No. RS-2024-00469482 & RS-2024-00509279).

## Impact Statement

Controllable generation methods such as FK steering can be applied to align language models with human preferences, including to improve their personalization or safety. Additionally, we show that FK steering can be used for automated red-teaming, which can inform model deployment. We recognize that any such method for controllable generation can be used to generate harmful samples by malicious actors. However, FK steering enables the research community to better understand properties of generative models and make them safer, which we believe will ultimately outweigh these harms.

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

# A. Text to Image Experiments

| Model | Params | **Base**$(k=1)$ | **Base**$(k=4)$ | **FK**$(k=4)$ | **FK**$(k=4, \text{parallel})$ |
|---|---|---|---|---|---|
| SD v1.4/v1.5 | 860M | 2.4s | 7.3s | 8.1s | 5.0s |
| SD v2.1 | 865M | 4.6s | 15.6s | 17.4s | 9.1s |
| SDXL | 2.6B | 11.5s | 42.3s | 43.5s | 21.7s |

*Table 7.* **Parameter counts and timing**. In this table, we provide inference timing for text-to-image diffusion models with FK steering. We include results for FK steering on a single NVIDIA-A100 GPU and a two-device parallel implementation. FK steering incurs only a minimal increase in time compared to independently generating $k$ particles. This gap shrinks as the diffusion model parameter count increases.

In this section, we explore the effect of $\lambda$ and the resampling schedule on particle diversity for text-to-image generation. Similar to Domingo-Enrich et al. (2024), we measure the diversity of generations using the CLIP (Radford et al., 2021) encoder $f_\theta$, so given $k$ $\{\mathbf{x}_0^i\}_{i=1}^k$ particles, we measure:

$$\text{CLIP-Div}\left(\{\mathbf{x}_0^i\}_{i=1}^k\right) := \sum_{i=1}^k \sum_{j=i}^k \frac{2}{k(k-1)} \left\| f_\theta(\mathbf{x}_0^i) - f_\theta(\mathbf{x}_0^j) \right\|_2^2. \tag{4}$$

Similar to section 4.1, we use the stable diffusion text-to-image models (Rombach et al., 2022) with the ImageReward human preference score (Xu et al., 2024) as the reward function. Here we use the difference potential.

We evaluate FK steering with different values of $\lambda$ and different resampling schedules, $[0, 20, 40, 60, 80]$ and $[0, 70, 75, 80, 85, 90]$. In table 8, we observe that for all values of $\lambda$ and the resampling schedule, the GenEval score of FK steering outperforms the base model. However, for lower values of $\lambda$, the CLIP diversity score is significantly higher, implying higher particle diversity. Similarly, in table 9, we observe that for higher values of $\lambda$, the human preference scores are higher, while the particle diversity is lower.

# B. Text Experiments

For all text experiments, we use publicly available SSD-LM[10], MDLM[11], and GPT2-Medium[12] checkpoints. For both text experiments, we generate sequences of length 50, conditioned on the prompts used by Han et al. (2023) to evaluate controllable text generation. We generate 20 continuations for each of the 15 prompts.

## B.1. Baselines

Following Han et al. (2023), for SSD-LM we iteratively generate these continuations in blocks of 25. Except for our $T = 5000$ quality experiment, we default to $T = 500$ for all SSD-LM experiments, and follow the multi-hot sampling procedure, with a top-p $= 0.20$ (Han et al., 2023). For toxicity gradient guidance, we set the learning rate $= 2000$. For MDLM, we condition on each prompt by prefilling the prompt tokens at inference time. The model is trained to generate tokens in blocks of 1024. For consistency, we only consider the first 50 tokens of each generated sample, after re-tokenizing with the SSD-LM tokenizer. We use 1000 steps for all MDLM experiments. For the GPT2-Medium baseline, we generate all samples with top-p $= 0.95$ and temperature $= 1.0$.

## B.2. FK steering Details

For all FK steering text experiments, we set $\lambda = 10.0$ and use the difference of rewards potential. We resample 50 times for each inference: at every 10 steps for SSD-LM and every 20 steps for MDLM. To convert intermediate SSD-LM states to text, we sample tokens from the logit estimate, $\widehat{\mathbf{x}}_t$, with top-p $= 0.20$. To convert intermediate MDLM states to text, we sample the masked tokens from the multinomial distribution given by $\widehat{\mathbf{x}}_t$. By default, we sample one intermediate text for SSD-LM, and four texts for MDLM. Rewards are averaged over these samples. For *Improved* FK steering with MDLM, we sample and evaluate 16 intermediate texts, rather than 4.

For *Improved* FK steering with SSD-LM, we take the more involved approach of fine-tuning the off-the-shelf toxicity classifier on intermediate states, $\widehat{x}_t$. To build a training dataset, we used reward toxicity classifier to identify 26K non-toxic

---

[10]https://huggingface.co/xhan77/ssdlm
[11]https://huggingface.co/kuleshov-group/mdlm-owt
[12]https://huggingface.co/openai-community/gpt2-medium

| Model | Sampler | Schedule | CLIP Div. | GenEval Score |
|---|---|---|---|---|
| SD v1.4 | FK($k = 4, \lambda = 10$) | 5-30-5 | 0.1437 | 0.4814 |
| SD v1.4 | FK($k = 4, \lambda = 10$) | 20-80-20 | 0.1050 | 0.5258 |
| SD v1.4 | FK($k = 4, \lambda = 2$) | 5-30-5 | 0.2321 | 0.4975 |
| SD v1.4 | FK($k = 4, \lambda = 2$) | 20-80-20 | 0.2239 | 0.4910 |
| SD v1.4 | base ($k = 4$) | - | 0.3158 | 0.4408 |
| SD v1.5 | FK($k = 4, \lambda = 10$) | 5-30-5 | 0.1459 | 0.4861 |
| SD v1.5 | FK($k = 4, \lambda = 10$) | 20-80-20 | 0.1038 | 0.5224 |
| SD v1.5 | FK($k = 4, \lambda = 2$) | 5-30-5 | 0.2330 | 0.4854 |
| SD v1.5 | FK($k = 4, \lambda = 2$) | 20-80-20 | 0.2252 | 0.5114 |
| SD v1.5 | base ($k = 4$) | - | 0.3115 | 0.4483 |
| SD v2.1 | FK($k = 4, \lambda = 10$) | 5-30-5 | 0.1259 | 0.5523 |
| SD v2.1 | FK($k = 4, \lambda = 10$) | 20-80-20 | 0.1061 | 0.5783 |
| SD v2.1 | FK($k = 4, \lambda = 2$) | 5-30-5 | 0.2051 | 0.5607 |
| SD v2.1 | FK($k = 4, \lambda = 2$) | 20-80-20 | 0.2213 | 0.5587 |
| SD v2.1 | base ($k = 4$) | - | 0.2948 | 0.5104 |
| SDXL | FK($k = 4, \lambda = 10$) | 5-30-5 | 0.1182 | 0.6056 |
| SDXL | FK($k = 4, \lambda = 10$) | 20-80-20 | 0.1055 | 0.6034 |
| SDXL | FK($k = 4, \lambda = 2$) | 5-30-5 | 0.1816 | 0.5863 |
| SDXL | FK($k = 4, \lambda = 2$) | 20-80-20 | 0.2111 | 0.5857 |
| SDXL | base ($k = 4$) | - | 0.2859 | 0.5571 |

*Table 8.* **Effect of $\lambda$ and resampling schedule on diversity.** Here we report *average* GenEval scores of all particles generation by FK steering to show that prompt fidelity increases for all particles. Moreover, we notice that lower values of $\lambda$ can also be used to generate diverse particles.

| Model | Sampler | Schedule | IR (Mean / Max) | HPS (Mean / Max) | CLIP Div. |
|---|---|---|---|---|---|
| SD v1.4 | base ($k = 4$) | - | 0.234 (0.800) | 0.245 (0.256) | 0.348 |
| SD v1.4 | FK ($k = 4, \lambda = 10.0$) | 5-30-5 | 0.506 (0.783) | 0.251 (0.255) | 0.193 |
| SD v1.4 | FK ($k = 4, \lambda = 10.0$) | 20-80-20 | 0.811 (0.927) | 0.258 (0.259) | 0.091 |
| SD v1.4 | FK ($k = 4, \lambda = 1.0$) | 20-80-20 | 0.502 (0.763) | 0.252 (0.256) | 0.173 |
| SD v1.4 | FK ($k = 4, \lambda = 1.0$) | 5-30-5 | 0.368 (0.723) | 0.248 (0.254) | 0.236 |
| SD v2.1 | base ($k = 4$) | - | 0.372 (0.888) | 0.253 (0.263) | 0.318 |
| SD v2.1 | FK ($k = 4, \lambda = 1.0$) | 5-30-5 | 0.582 (0.835) | 0.258 (0.261) | 0.180 |
| SD v2.1 | FK ($k = 4, \lambda = 10.0$) | 20-80-20 | 0.891 (1.006) | 0.264 (0.266) | 0.087 |
| SD v2.1 | FK ($k = 4, \lambda = 1.0$) | 20-80-20 | 0.579 (0.826) | 0.257 (0.261) | 0.164 |
| SDXL | base ($k = 4$) | - | 0.871 (1.236) | 0.289 (0.296) | 0.248 |
| SDXL | FK ($k = 4, \lambda = 10.0$) | 5-30-5 | 1.032 (1.186) | 0.293 (0.295) | 0.123 |
| SDXL | FK ($k = 4, \lambda = 10.0$) | 20-80-20 | 1.211 (1.298) | 0.296 (0.297) | 0.071 |

*Table 9.* **Effect of $\lambda$ and resampling schedule on diversity.** Here we report the *average* ImageReward and HPS scores of all particles generation by FK steering to show that sample quality increases for all particles.

and 26K toxic texts from the OpenWebText corpus (Gokaslan et al., 2019). We then applied the SSD-LM forward process $q$ to noise the text to random timestep $t$, and then use the base model to infer $\hat{x}_t$. We then fine-tune the off-the-shelf reward classifier to estimate the toxicity probability of the original text given the intermediate text.

We fine-tune three reward models for different SSD-LM time-step ranges:

$$t \in [500, 300), [300, 200), [200, 100)$$

We train with batch size $= 16$ and learning rate $= 5e - 7$, using a constant learning rate with 50 warm-up steps. We train with cross entropy loss, and use a weighting (0.99 non-toxic, 0.01 toxic), due to the rarity of toxicity in the original data distribution. For the gradient-based guidance baseline for SSD-LM, we use the default guidance scale from Han et al. (2023)[13].

## C. Consistency of Particle Approximations

In this section, we prove that using SMC with multinomial resampling leads to a consistent approximation of the target distribution, that is, suppose we have $k$ particles $\mathbf{x}_0^i$ and potentials $G_0(\mathbf{x}_T^i, \ldots, \mathbf{x}_0^i)$, then the weighted empirical distribution converges to the target $p_{\text{target}}(\mathbf{x}_0) \propto p_\theta(\mathbf{x}_0) \exp(\lambda r(\mathbf{x}_0))$. Let $\mathbf{w}_t^i$ denote the normalized potential scores

$$\mathbf{w}_t^i := \frac{1}{\sum_{j=1}^k G_t(\mathbf{z}_t^j)} G_t(\mathbf{z}_t^i) \tag{5}$$

where $\mathbf{z}_t^i = (\mathbf{x}_T^i, \ldots, \mathbf{x}_t^i)$ for $i \in \{1, \ldots, k\}$ denotes the path sampled till time $t$. Then we show that as $k \to \infty$:

$$\sum_{i=1}^k \mathbf{w}_0^i \delta_{\mathbf{z}_0^i} \Rightarrow \frac{1}{\mathbf{Z}} p_\theta(\mathbf{x}_T, \ldots, \mathbf{x}_0) \exp(\lambda r(\mathbf{x}_0)) \tag{6}$$

which implies that $\sum_{i=1}^k \mathbf{w}_0^i \delta_{\mathbf{x}_0^i} \Rightarrow p_{\text{target}}(\mathbf{x}_0)$.

The proof of consistency relies on the following two facts:

- The process on the extended space $\mathbf{z}_t$ is also Markov, that is $p_\theta(\mathbf{z}_t \mid \mathbf{z}_{t+1}, \ldots, \mathbf{z}_T) = p_\theta(\mathbf{z}_t \mid \mathbf{z}_{t+1})$.

- For each $t \in \{T-1, \ldots, 0\}$, the particle-based approximation is consistent, so

$$\sum_{i=1}^k \mathbf{w}_t^i \delta_{\mathbf{z}_t^i} \Rightarrow p_{\text{FK},t}(\mathbf{x}_T, \ldots, \mathbf{x}_t). \tag{7}$$

Note that, since $\prod_{t=T}^0 G_t = \exp(\lambda r(\mathbf{x}_0))$, eq. (7) for $t = 0$ implies eq. (6).

To prove consistency we rely on lemma 11.1 in Chopin et al. (2020) which proves weak convergence of the SMC particle approximations.

**Lemma 1** (Lemma 11.1 in Chopin et al. (2020)). *Suppose the potential functions $\{G_t\}_{t=T}^0$ are upper-bounded and $\mathbf{x}_t \in \mathbf{R}^d$, then for all $t \in \{T, \ldots, 0\}$, there exists a constant $c_t > 0$ such that for all continuous and bounded functions $\phi : \mathbf{R}^{d \times t} \to \mathbf{R}$, for all $k$ we have:*

$$\mathbb{E}\left[\left|\sum_{i=1}^k \mathbf{w}_t^i \phi(\mathbf{z}_t^i) - \mathbb{E}_{p_{\text{FK},t}}[\phi(\mathbf{z}_t)]\right|^2\right] \leq c_t \frac{1}{k} \|\phi\|_\infty^2 \tag{8}$$

*where $\mathbf{w}_t^i = \frac{G_t(\mathbf{z}_t^i)}{\sum_{j=1}^k G_t(\mathbf{z}_t^j)}$ are the normalized resampling weights.*

Lemma 1 implies that the weighted empirical distribution for all $t$ are consistent, proving eq. (7). Now, note that eq. (8) implies that the weighted empirical distribution $\sum_{i=1}^k \mathbf{w}_0^i \delta_{\mathbf{x}_0^i}$ converges to $p_{\text{target}}(\mathbf{x}_0)$, since for all continuous and bounded functions $\psi : \mathbf{R}^d \to \mathbf{R}$, eq. (8) implies that

$$\mathbb{E}\left[\left|\sum_{i=1}^k \mathbf{w}_0^i \psi(\mathbf{x}_0^i) - \mathbb{E}_{p_{\text{FK},0}}[\psi(\mathbf{x}_0)]\right|^2\right] \leq c_0 \frac{1}{k} \|\psi\|_\infty^2 \tag{9}$$

therefore, for $t = 0$ the particle-approximation converges to the target distribution:

$$\sum_{i=1}^k \mathbf{w}_0^i \delta_{\mathbf{x}_0^i} \Rightarrow \frac{1}{\mathbf{Z}} p_\theta(\mathbf{x}_0) \exp(\lambda r(\mathbf{x}_0)) \tag{10}$$

---

[13]Provided in private communication by the authors of Han et al. (2023).

## D. Feynman-Kac IPS Discussion

### D.1. Choice of proposal distribution

Here we discuss various choices for twisting the transition kernel towards high reward samples:

- **Gradient-based guidance**: For continuous-state models and differentiable rewards, we can use gradient's from the reward (Sohl-Dickstein et al., 2015; Song et al., 2020b; Bansal et al., 2023; Wu et al., 2023a) to guide the sampling process. Suppose $p_\theta(\mathbf{x}_t \mid \mathbf{x}_{t+1}, \mathbf{c}) = \mathcal{N}(\mu_\theta(\mathbf{x}_t, \mathbf{c}), \sigma_\theta^2 I_d)$, then we can *twist* the transition kernel using reward gradients:

$$\mathcal{N}\left(\mu_\theta(\mathbf{x}_t, \mathbf{c}) + \sigma_\theta^2 \lambda \nabla_{\mathbf{x}_t} r_\phi(\mathbf{x}_t, \mathbf{c}), \sigma_\theta^2\right), \tag{11}$$

  where $r_\phi$ is the intermediate reward, either learned or evaluated at the reward on the denoised state $r(\mathbf{x}_0 = \widehat{\mathbf{x}}_t)$.

- **Discrete normalization**: For discrete diffusion models, such as masked diffusion language model (MDLM) (Sahoo et al., 2024; Shi et al., 2024), we can also estimate the normalization constant:

$$\sum_{\mathbf{x}_t} p_\theta(\mathbf{x}_t \mid \mathbf{x}_{t+1}, \mathbf{c}) G_t(\mathbf{x}_T, \ldots, \mathbf{x}_t, \mathbf{c}) \tag{12}$$

  and sample from $p_{\text{FK},t}(\mathbf{x}_t \mid \mathbf{x}_{t+1}, \ldots, \mathbf{x}_T) \propto p_\theta(\mathbf{x}_t \mid \mathbf{x}_{t+1}) G_t(\mathbf{x}_T, \ldots, \mathbf{x}_t)$.

However, such methods for twisting the transition kernel can lead to increased sampling time compared to sampling from the *base* model $p_\theta$.

### D.2. How existing work fits into FK steering

TDS (Wu et al., 2023a) uses SMC to do conditional sampling with a marginally trained model and a differentiable reward. They make the choices:

- **Potential.** $G_t(\mathbf{x}_t, \mathbf{x}_{t+1}) = \exp(\lambda(r(\mathbf{x}_t) - r(\mathbf{x}_{t+1})))$, where the reward is computed on the denoised state $r(\mathbf{x}_t) = r(\mathbf{x}_0 = \widehat{\mathbf{x}}_t)$.

- **Proposal generator.** They use classifier-guidance to approximate the conditional score model $s_\theta(\mathbf{x}_t, t, y) \approx s_\theta(\mathbf{x}_t, t) + \nabla_{\mathbf{x}_t} \log p_\theta(y \mid \mathbf{x}_0 = \widehat{\mathbf{x}}_t(\mathbf{x}_t, t))$ and use the following proposal generator $\tau(\mathbf{x}_t \mid \mathbf{x}_{t+1})$:

$$\tau(\mathbf{x}_t \mid \mathbf{x}_{t+1}) = \mathrm{N}(\Delta t[f - gg^\top s_\theta(\mathbf{x}_t, t, y)], g(t)\Delta t) \tag{13}$$

FK steering allows for a more flexible use of potentials $G_t$, as well as proposal generators. For instance, Nichol et al. (2021) show that conditionally trained scores outperform classifier-guidance even when the classifier is trained on noisy states $\mathbf{x}_t$. However, as shown by Ghosh et al. (2024), conditionally trained models still have failure modes. Therefore, we demonstrate how particle based methods can be used to improve the performance of conditionally trained models as well. Furthermore, FK steering allows these methods to be applied to discrete-space diffusions as well as non-differentiable rewards.

Soft value-based decoding in diffusion models (SVDD) is another particle-based method. Instead of using SMC, SVDD utilizes a nested importance sampling algorithm (see algorithm 5 of Naesseth et al. (2019)) for the proposal generator with a single particle. SVDD makes the following choices:

- **Potential.** Similar to TDS, they use the potential $G_t = \exp(\lambda(r(\mathbf{x}_t) - r(\mathbf{x}_{t+1})))$ where $r(\mathbf{x}_t)$ can be off-the-shelf like TDS or learned from model samples.

- **Sampler.** SVDD uses the base model as the proposal generator and generates $k$ samples at each step, selects a *single sample* using importance sampling and makes $k$ copies of it for the next step.

With $\lambda = \infty$, SVDD is equivalent to doing best-of-$n$ at each step, since the authors recommend sampling from $p_{\text{target}}(\mathbf{x}_0) \propto \lim_{\lambda \to \infty} p_\theta(\mathbf{x}_0) \exp(\lambda r(\mathbf{x}_0))$. We note that similar to SVDD, $p_{\text{FK},0}$ can be sampled using nested importance sampling.

### D.3. Adaptive Resampling

Following Naesseth et al. (2019); Wu et al. (2023a), we can use adaptive resampling to increase diversity of samples. Given $k$ particles $\mathbf{x}_t^i$ and their potentials $G_t^i$, we define the effective sample size (ESS):

$$\text{ESS}_t = \frac{1}{\sum_{i=1}^k \left(\widehat{G}_t^i\right)^2} \tag{14}$$

where $\widehat{G}$ refers to the normalized potentials and $\text{ESS}_t \in [1, k]$. If $\text{ESS}_t < \frac{k}{2}$, then we skip the resampling step. This encourages particle diversity.

## E. FK steering samples

In this section, we show the effect of various sampling parameters, such as potentials, the temperature parameter $\lambda$, number of sampling steps, etc. on the diversity of samples. We use the stable diffusion XL-base (SDXL) as the base model and proposal generator and the ImageReward (Xu et al., 2024) human preference score model as the reward function. We also use adaptive resampling introduced in appendix D.3. We compare FK steering against generating $k$ independent samples, using the same seed for generation, thus providing a counterfactual generation.

- **Effect of $\lambda$**: The parameter $\lambda$ is used to define the target distribution:

$$p_{\text{target}}(\mathbf{x}_0) = \frac{1}{\mathbf{Z}} p_\theta(\mathbf{x}_0) \exp(\lambda r(\mathbf{x}_0)), \tag{15}$$

therefore, higher values of $\lambda$ upweight higher reward samples $\mathbf{x}_0$. Similarly, the potentials also use $\lambda$ which affects resampling. We generate $k = 4$ samples from the SDXL using FK steering as well as $k = 4$ independent samples using the max potential. In fig. 7, we observe that using FK steering improves prompt fidelity, and higher values of $\lambda$ improve fidelity at the cost of particle diversity.

- **Effect of potential**: In fig. 6, we observe that FK steering with the max potential reduces diversity compared to the difference potential. Here we use $\lambda = 2$ and generate $k = 8$ samples using the max and difference potential.

- **Effect of sampling steps.** In fig. 6, we observe that diversity can be increased by increasing the number of sampling steps from 100 to 200. Here we use $[180, 160, 140, 120, 0]$ and $[80, 60, 40, 20, 0]$ as the resampling interval. We note that even if the samples $\mathbf{x}_0$ share the same particle as parent, there is diversity in the final samples.

- **Effect of interval resampling**: In fig. 8, we show that using interval resampling even with 100 sampling steps produces diversity in samples. For comparison, see fig. 8 for the independent versus FK steering generations.

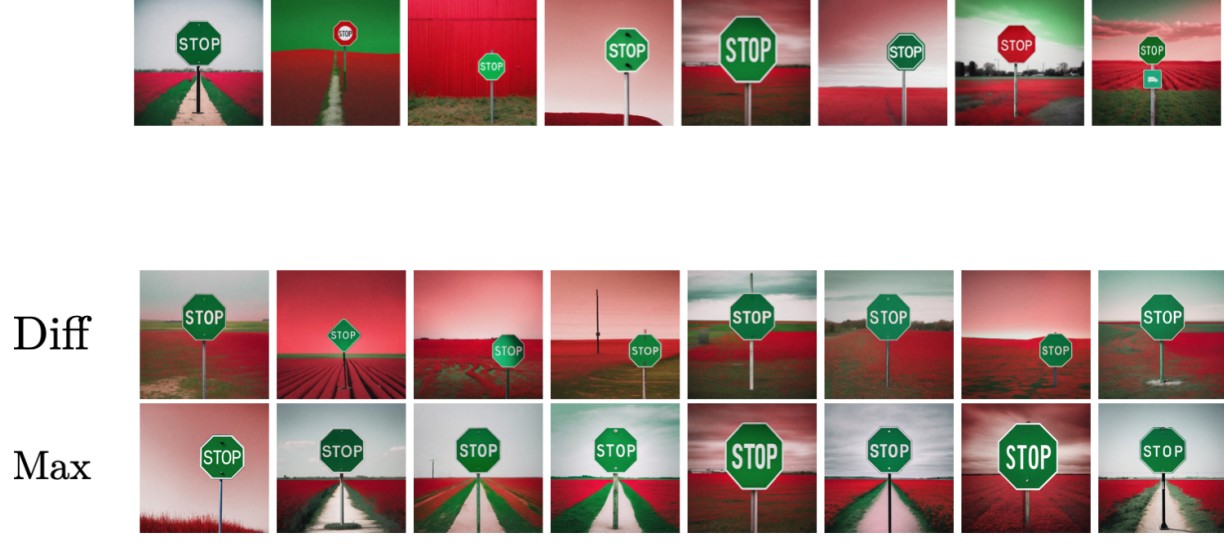

*Figure 6.* **Max versus Difference potential**: In the top row, we plot 8 independent samples from the base model and in the bottom two rows, we have the FK steering particles for the max and difference potentials. Using the max potential reduces diversity compared to the difference potential. However, we note that by increasing the number of sampling steps, the diversity of the samples can be increased. Caption: *a green stop sign in a red field*

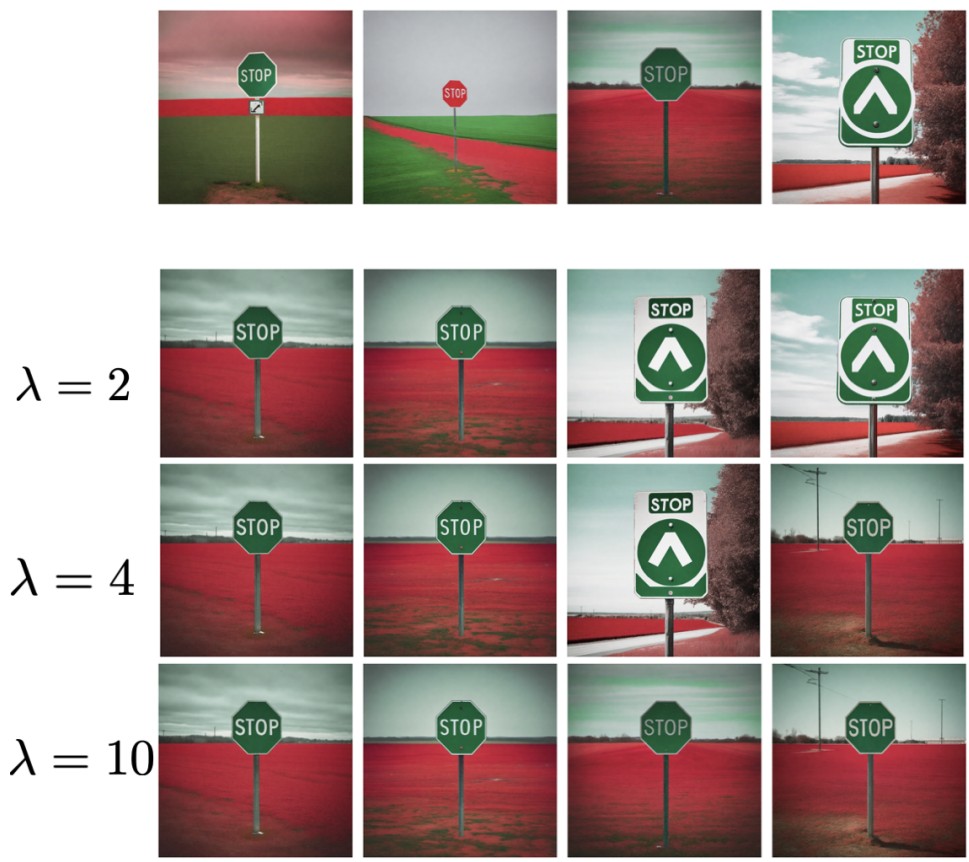

*Figure 7.* **Effect on $\lambda$ on diversity:** In the top panel, we plot 4 independent samples from the base model and in the bottom 3 panels, we have the FK steering particles for varying values of $\lambda$. We observe that increasing $\lambda$ leads to a decrease in diversity, at the cost of higher prompt fidelity and improved aesthetic quality, compared to the first row which has 4 independent samples. Caption: *a green stop sign in a red field*

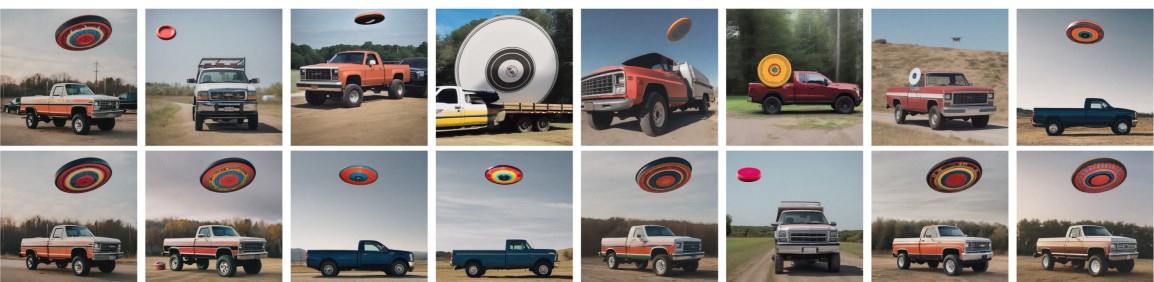

*Figure 8.* **Increased prompt fidelity:** In this generation, we compare $k = 8$ independent samples (top panel) versus $k = 8$ samples from FK steering (bottom panel). FK steering selects samples which follow the prompt. Caption: *a photo of a frisbee above a truck*

