# OpenReview forum: "A General Framework for Inference-time Scaling and Steering of Diffusion Models"
_ICML.cc/2025/Conference — ICML 2025 poster_

### Official Review · Reviewer_FaZn · 2025-02-21

**Overall Recommendation:** 4

**Summary:**

This study proposes FK steering, a scalable inference method for both image and text diffusion models. FK steering consists of a proposal generator, a potential function, and an intermediate reward. In each denoising interval, the diffusion model generates multiple proposals, and the potential function uses one of three functions—difference, max, or sum—to reweight the particles for resampling in the next denoising step. The intermediate reward employed by the potential function is chosen from among expected x_0, many sample reward, or learned reward. FK steering is a training-free method that outperforms fine-tuning methods like DPO. It even surpasses SDXL (with 2.6B parameters) when using SDv2.1 (with 0.8B parameters), and its superiority is evaluated on text-to-image tasks using metrics such as GenEval, image reward, and HPS. Additionally, the method demonstrates effectiveness in text diffusion models, proving that it is a general framework.

**Claims And Evidence:**

Overall, the experiments are well-executed and provide strong support for most of the claims. However, a few questions remain, and additional evidence would benefit the paper. It would be beneficial to strengthen experiments in the text-to-image domain with up-to-date open-weight models. Can FK steering make a Stable Diffusion 3 medium (2B) [A] model outperform the large (8B) [B] model? Or if it can surpass advanced models like Flux-dev [C]. Can FK steering make a timestep-distilled model like Flux-Schnell [D] outperform Flux-dev? In addition, experiments on ImageNet in Appendix A would be more convincing if they included additional metrics like FID and IS. Given that classifier guidance by Nichol & Dhariwal (2021) significantly improves these metrics, it is unclear if FK steering can achieve similar improvements.

[A] Stability AI, https://huggingface.co/stabilityai/stable-diffusion-3.5-medium

[B] Stability AI, https://huggingface.co/stabilityai/stable-diffusion-3.5-large

[C] Black Forest Labs, https://huggingface.co/black-forest-labs/FLUX.1-dev

[D] Black Forest Labs, https://huggingface.co/black-forest-labs/FLUX.1-schnell

**Essential References Not Discussed:**

No critical issues were found.

**Experimental Designs Or Analyses:**

Overall, the experiments effectively demonstrate the benefits of FK steering. However, I am curious as to why the GenEval performance does not match between Table 1 and Table 3. Additionally, for text-to-image tasks, it would be helpful to include more qualitative comparisons using multiple seeds.

**Methods And Evaluation Criteria:**

The proposed methods make sense for the inference time scaling. The paper clearly explains its components—a proposal generator, potential function, and intermediate rewards—along with all the possible choices for each, which are well-motivated. The evaluation criteria are also appropriate; for text-to-image tasks, metrics like GenEval and HPS, which are commonly used in the field, are employed. However, I have less expertise in text diffusion model experiments, so insights from other experts would be valuable.

**Other Comments Or Suggestions:**

It would be beneficial to include a table or plot comparing the increase in inference time and memory cost. Additionally, showing whether performance continues to improve when increasing the number of particles beyond k=8 would help demonstrate the scalability of the proposed method.

**Other Strengths And Weaknesses:**

As mentioned previously, the key strength of this paper is that it achieves effective results without relying on finetuning or gradient guidance. It explains each option and validates their effectiveness through ablation studies. However, the evaluation could be improved by incorporating experiments on more up-to-date models and enhancing the metrics in the ImageNet evaluations.

**Questions For Authors:**

In the early denoising steps, the expected x_0 will be blurry as the diffusion models predict the “expectation” at each denoising step. So, the reward model may not be able to provide a reliable signal at that point. How does the reward model play a meaningful role in the earlier denoising steps?

**Relation To Broader Scientific Literature:**

Previous work aimed at improving diffusion model quality, text alignment, and human preference has largely relied on fine-tuning methods or inference steering techniques that employ gradient guidance from reward models. In contrast, this paper is novel and effective in that it is training-free and bypasses the need for gradient guidance.

**Theoretical Claims:**

I reviewed the equations and theoretical claims presented in the main manuscript. The notations are meticulously defined, and the derivations appear to be correct.

---

> ### Author Rebuttal · Authors · 2025-04-01
>
> Thank you for highlighting the strength of our results. We also deeply appreciate your comments about the high-quality execution of our paper, the rigor/strength of our evaluations, and the clarity of our exposition.
>
> _Can FK steering make a Stable Diffusion 3 medium (2B) [A] model outperform the large (8B) [B] model?_
>
> **SD3 2B + FK steering indeed outperforms SD3 8B.** Stable Diffusion 3 2B with FK steering using just $k=3$ particles achieves a higher prompt fidelity score than the 8B parameter model.
>
> | Model                  | Param. Count | GenEval score    |
> |-|-|-|
> | SD3 medium                 | 2B  | 0.72 |
> | SD3 medium + FK(k=3)  | 2B    | **0.761** |
> | SD3 large   | 8B   | 0.74 |
>
> _Can FK steering make a timestep-distilled model like Flux-Schnell outperform Flux-dev?_
>
> **Distillation + FK steering**. Improving timestep distillation is an interesting future direction and a natural extension of FK steering.
>
> _Given that classifier guidance by Nichol & Dhariwal (2021) significantly improves these metrics, it is unclear if FK steering can achieve similar improvements._
>
> **Combining gradients with FK steering**. Our ImageNet experiment show that FK steering can improve on classifier-guidance. We have also added ImageReward gradient-guidance results:
>
> | Model                  | GenEval | IR    | HPS   | Sampling Time  |
> |-|-|-|-|-|
> | SDv1.5                 | 0.44    | 0.187 | 0.245 | 2.4s  |
> | SDv1.5 + IR guidance   | 0.450    | 0.668 | 0.245 | 20s   |
> | SD v1.5 + FK (k = 4)   | **0.54**    | **0.898** | **0.263** | 8.1s  |
> | SD v1.5 + IR guidance + FK (k = 4)   | **0.56**    | **1.290** | **0.268** | 55s |
>
> Here, FK steering **both outperforms and improves gradient guidance**.
>
> _I am curious as to why the GenEval performance does not match between Table 1 and Table 3. Additionally, for text-to-image tasks, it would be helpful to include more qualitative comparisons using multiple seeds._
>
> - **Average vs best particle performance**. Table 1 has results from the best particle out of $k$ while Table 3 has results averaged over all $k$ particles. We do this to highlight that FK steering improves sample quality and prompt alignment metrics for all the particles. We will make this distinction more clear in the text and captions for the tables in the revised draft.
>
> _I'm interested in knowing whether the proposed method offers an orthogonal contribution to classifier-free guidance, or if it only proves effective at a specific classifier-free guidance scale._
>
> - Thanks for this question. We ran FK steering with $k=4$ with a lower guidance scale in the table below. However,  we note that higher guidance scales lead to better performance.
>
> | Model | Scale |  GenEval   | HPS   |IR   |
> |-|-|-|-|-|
> | SDv1.5  + FK | 4    | 0.52 | 0.795 | 0.256  |
> | SDv1.5  + FK   | 7.5   | 0.54 | 0.898 | 0.263  |
> | SDv2.1  + FK  | 4 | 0.59 | 0.901 | 0.262 |
> | SDv2.1  + FK  | 7.5 | 0.62 | 1.006 | 0.268  |
> | SDXL  + FK| 4    | 0.62 | 1.264 | 0.299  |
> | SDXL  + FK| 7.5    | 0.64 | 1.298 | 0.302  |
>
> _The evaluation could be improved by incorporating experiments on more up-to-date models and enhancing the metrics in the ImageNet evaluations._
>
> - **ImageNet**. We will add these numbers in the final paper revision.
> - **Up-to-date models**. We have included additional SD3 results above.
>
> _It would be beneficial to include a table or plot comparing the increase in inference time and memory cost_
>
> - **Memory and sampling time increase**. In table 6, we show the increase in sampling time from FK steering with $k=4$ particles. We include sample times for a single GPU run as well as parallel generation with 2 GPUs. We will update our draft to contain FLOP/memory details.
>
> _Showing whether performance continues to improve when increasing the number of particles beyond k=8 would help demonstrate the scalability of the proposed method._
>
> - **Beyond k=8**. In figure 4, we include the GenEval and ImageReward scores for increasing values of $k$, from 2 to 16. The figure indicates that increasing particle count improves both scores. For our text experiments, we will include additional results in the revised version of the paper.
>
> _In the early denoising steps, the expected x_0 will be blurry as the diffusion models predict the “expectation” at each denoising step...How does the reward model play a meaningful role in the earlier denoising steps?_
>
> Thank you for this question. In figure 5 in the appendix, we plot the correlation of the reward $r_\phi(x_t) = r(x_0 = E_\theta[x_0 | x_t])$ with the reward at the terminal step $r(x_0)$. For ImageReward, we see a high correlation early in the generation. We demonstrate that learning the rewards will improve this correlation even further and can lead to improved performance. For instance, for toxicity we see a low correlation, prompting us to learn the intermediate rewards, which improves the generation's attribute accuracy as shown in table 4, see SSD-LM with learned $r_\phi$.

---

> > ### Comment · Reviewer_FaZn · 2025-04-02
> >
> > I appreciate the authors' rebuttal, which addresses my questions. I also read the reviews from the other reviewers and noted that there are concerns regarding the novelty of the work. However, I believe the strong empirical performance demonstrated in the paper outweighs these concerns. I will maintain my current score.

---

> > > ### Author Response · Authors · 2025-04-02
> > >
> > > Thank you for reading our response and highlighting the approach’s strong results.

---

### Official Review · Reviewer_9NEc · 2025-02-25

**Overall Recommendation:** 3

**Summary:**

This paper proposes Feynman-Kac steering for diffusion models using pretrained reward functions like ImageReward.

**Update after rebuttal**

My main concern was that earlier works proposing FK for diffusion were not clearly acknowledged and discussed. During the rebuttal phase the authors proposed specific revisions to remedy this. Therefore I have increased my score.

**Claims And Evidence:**

Yes

**Essential References Not Discussed:**

"Conditional sampling within generative diffusion models" (https://arxiv.org/abs/2409.09650v1) (see Relation To Broader Scientific Literature)

"Nested Diffusion Processes for Anytime Image Generation" (https://arxiv.org/abs/2305.19066) -- a similar idea to the "many-sample" approach mentioned on L246 was discussed in Section 5 of this reference.

**Experimental Designs Or Analyses:**

Did not check.

**Methods And Evaluation Criteria:**

Yes

**Other Comments Or Suggestions:**

N/A

**Other Strengths And Weaknesses:**

The writing is clear and the experiments are nice. Although I feel that the technical contributions is somewhat limited (see comments on prior work and missing references) I do appreciate the clarity of the exposition compared to earlier papers which are harder to digest. Using pretrained rewards is a good idea, and the experiments showcase clear practical applications of the method. However, I feel that earlier works proposing FK steering of diffusion models should be more clearly acknowledged.

**Questions For Authors:**

Please see Relation To Broader Scientific Literature and Essential References Not Discussed.

**Relation To Broader Scientific Literature:**

I have some concerns about the novelty claims in this paper in relation to prior work. It is certainly not the first to propose FK for steering diffusion models. I feel that is it too strong to say things like "we propose FK steering" as on L17 of abstract. To me, the main contribution of the work is in making a clear and experimentally-demonstrated connection to modern text-to-image settings and the idea of using pretrained reward functions.

"Conditional sampling within generative diffusion models" (https://arxiv.org/abs/2409.09650v1). An earlier work discussing FK steering of diffusion models. I think this one should be cited and discussed.

This paper does cite and compare to the paper "Derivative-Free Guidance in Continuous and Discrete Diffusion Models with Soft Value-Based Decoding" (SVDD) (https://arxiv.org/abs/2408.08252). However, on L868 this paper says that SVDD only selects a single sample, but that is only the case for the choice alpha=0, whereas for alpha > 0 SVDD seems very similar (if not identical?) to FK (please clarify if this is incorrect).

Some other concurrent related works (just an FYI):
"Composition and Control with Distilled Energy Diffusion Models and Sequential Monte Carlo" (https://arxiv.org/abs/2502.12786). Also uses FK steering but requires an energy-based model (EBM) (which they distill from a diffusion model) to build reward functions that depend on the density p_t (for example, temperature control and composition). It might be nice to highlight how pretrained reward functions as in you suggest help avoid the need for EBMs.

"Debiasing Guidance for Discrete Diffusion with Sequential Monte Carlo" (https://arxiv.org/abs/2502.06079). Similar approach for discrete temperature control.

**Theoretical Claims:**

I believe there was only proof in Appendix E, which is a direct application of a lemma of Chopin et al. It appears correct to me.

---

> ### Author Rebuttal · Authors · 2025-04-01
>
> Thank you for highlighting the quality of our writing and experiments, and the practical applications of our approach.
>
> _Concerns regarding novelty claims in this paper._
>
> Various fields such as statistics, posterior inference [Naesseth et al 2019], have used Feynman-Kac interacting particle systems for rare-event sampling and posterior inference with state-space models. We evaluate using FK-IPS to sample from target distributions defined by arbitrary rewards and diffusion models.
>
> FK-IPS requires two things, (1) potential functions used to up-weight sample paths that yield high rewards, see eq 2, and (2) a method for sampling from the FK distributions. In this work, we provide:
>  - **New potentials**.
>      1. We identify a simple condition in equation 3  that potentials must satisfy to consistently estimate the target distribution, and show that various new choices of potentials are possible.
>      2. The traditionally used DIFFERENCE potential minimizes the variance of the potentials (see thm 10.1 in [Chopin et al., 2020]), not necessarily generating high-reward samples.
>         - For bounded rewards (e.g. ImageReward), using the DIFFERENCE potential can paradoxically eliminate particles that achieve the maximum reward early in generation.
>         - In tables 3 & 5, we show that a potential we develop that satisfies the product conditions, yields higher reward samples than the difference potential.
>     3. We show that diffusion models offer many choices of intermediate rewards with different trade-offs between compute and knowledge of the terminal sample's reward. We propose two novel choices:
>         - *Learned from data*. Using data and the noising process, we train intermediate rewards using regression. This objective generalizes learning a classifier trained on noisy states in Dhariwal et al., 2023 to arbitrary reward functions.
>         - *Many-sample intermediate reward*. Using samples from $p_\theta(x_0 \mid x_t)$ we define intermediate rewards of the form: $\log \frac{1}{N} \sum_{i=1}^N \exp(r(x_0^i))$.
> - **Sampling**.
>    1. We show the effectiveness of the base model as the proposal generator, which is faster than techniques like gradient guidance and expands the use of SMC to discrete-state diffusion models and non-differentiable rewards.
>
> With this framework, we show **that inference-time steering of diffusion models is remarkably effective.**  FK steering:
> - **Outperforms fine-tuning**. With only $2$ particles, FK steering outperforms DPO, DDPO fine-tuning approaches on prompt fidelity metrics for all the text-to-image models considered.
> - **Outperforms gradient-guidance**. FK steering outperforms gradient guidance and is **significantly faster** (8.1s vs 20s for SDv1.5).
> - **Gradient-free control of discrete models**: FK steering can be gradient-free and enables plug-and-play control of discrete-space diffusion models.
> - **Overcomes parameter counts**. FK steering enables smaller models (0.8B) to outperform much larger models (2.6B) with less total compute.
>
> _This paper says that SVDD only selects a single sample, but that is only the case for the choice alpha=0, whereas for alpha > 0 SVDD seems very similar (if not identical?) to FK_
>
> **Comparison with SVDD**. SVDD selects a **single proposal** by sampling from a categorical distribution over the proposals $x_t^i$, with probability proportional to $\exp(r(x_t^i)/ \alpha)$  (see line 4 in algorithm 1). SVDD is a specific instantiation of nested importance sampling (algorithm 5 in Naesseth et al. 2019). FK steering uses SMC, which selects $k$ proposals instead of one, However, as noted in lines 198-204, it can use other particle-based samplers.
>
> **Prior works**.
>  1. **Nested diffusion sampling**. Elata et al 2023 uses an inner denoising loop to sample a single $x_0$ for each state $x_t$, rather than using $E_{\theta}[x_0 | x_t]$, in the transition kernel. This approach samples from the diffusion many times, but does not retain multiple samples for each step $x_t$.
>
>   2. **Conditional Sampling within Generative Diffusion Models** We discuss the relevant method from this review paper, Wu et al 2023, which proposes a particle-based sampler for conditional sampling with continuous diffusion models with $\log p(y | x)$ as reward. In contrast, we steer with arbitrary rewards and present various choices of potentials.
>
> Thank you for highlighting these references, we will add include them in the revised draft.
>
> The other related works cited were posted after the ICML submission deadline. Due to limited space, we will add a discussion on these works in either the final draft or in the discussion period.
>
> ### References
>
> [Wu et al 2023] Wu, Luhuan, et al. "Practical and asymptotically exact conditional sampling in diffusion models." (2023)
>
> [Chopin et al., 2020] Chopin, Nicloas et al. "An Introduction to Sequential Monte Carlo" (2020).
>
> [Naesseth et al. 2019] Naesseth, Christian et al. "Elements of Sequential Monte Carlo." (2019).

---

> > ### Comment · Reviewer_9NEc · 2025-04-03
> >
> > I want to clarify that I appreciate the contributions of the paper -- my concerns are about the writing seeming to imply being the first to propose applying FK to diffusion (for exampe, L17 of abstract, L37). To me, the main contributions of the work are in making a clear connection to modern text-to-image settings, proposing new reward functions, and providing strong experimental support — all of which I appreciate. I just feel that earlier works proposing FK for diffusion should be more clearly acknowledged — I would consider raising my score if the authors could propose some concrete changes to do so.
> >
> >
> > UPDATE: (I had posted this as an Official Comment but realized that is not visible to the authors). I appreciate the revisions and have increased my score.

---

> > > ### Author Response · Authors · 2025-04-04
> > >
> > > Thank you for engaging with our response. We propose the following concrete revisions:
> > >
> > > 1. L17
> > >  - *Original*: "In this work, we propose Feynman-Kac (FK) steering, a framework for inference-time steering diffusion models with reward functions."
> > >  - *Revised*: "We apply Feynman-Kac (FK) interacting particle systems to the inference-time steering of diffusion models with arbitrary reward functions, which we refer to as _FK steering_."
> > >
> > >  2. L37
> > >  - *Original*: "We introduce Feynman-Kac (FK) steering, a framework for inference-time steering of diffusion models.
> > >
> > >   - *Revised*: "Feynman-Kac measures provide a flexible framework for conditional sampling. FK measures have been used with interacting particle systems (Trippe et al., 2022, Wu et al., 2023a,  2024, Zhao et al., 2024) and divide and conquer approaches (Janati et al., Zhao et al., 2024) to enhance conditional sampling with diffusion models. In this work, we show that FK-IPS methods can provide a general framework for steering diffusion-based generative models with arbitrary rewards, which we refer to as FK steering."
> > >
> > >
> > >
> > > 3. L158
> > > - *Original*: "FK steering builds on top of recent work such as TDS (Wu et al., 2023a) and others (Trippe et al., 2022; Cardoso et al., 2023; Dou & Song, 2024) that propose particle-based methods for sampling from unnormalized distributions. In appendix F.2, we show how TDS (Wu et al., 2023a) and SVDD (Li et al., 2024) are examples of FK-IPS (Moral, 2004). Our experiments demonstrate that expanding the choice of potentials, rewards, and samplers provides several improvements, such as higher-reward samples."
> > >
> > > - *Revised*: "FK steering builds on top of recent works that sample from Feynman-Kac path distributions for conditional sampling with diffusion models, either using particle-based sampling (Trippe et al., 2022, Wu et al., 2023a, Cardoso et al., 2023, Dou & Song, 2024, **Zhao et al., 2024**) or gradient-based sampling (**Chung et al., 2022, Janati et al., 2024**). In appendix F.2, we show how TDS (Wu et al., 2023a) and SVDD (Li et al., 2024) are examples of FK-IPS (Moral, 2004). Our experiments demonstrate the effectiveness of these methods for new settings, and the value of expanding the choice of potentials, rewards, and samplers.
> > >
> > >
> > > We are happy to iterate on these revisions to address any remaining concerns.

---

### Official Review · Reviewer_49Va · 2025-03-09

**Overall Recommendation:** 2

**Summary:**

This paper proposes a general framework for inference-time scaling and steering of diffusion models using Feynman-Kac (FK) particle resampling. It claims to unify various existing steering methods by casting them into a single FK-IPS (Feynman-Kac Interacting Particle System) framework. The framework is validated empirically with relatively small numbers of particles.

**Claims And Evidence:**

The main claim—unifying inference-time steering methods into a single FK-based framework— might lack sufficient novelty. Specifically, the DIFFERENCE potential is essentially the common weighting strategy previously utilized by inference-time steering approaches such as SMC; the other two variants are also not too impressive. Even Figure 1 which shows the high-level principles does not appear to be significantly different as compared to current inference-time diffusion model guidance/alignment papers.

**Essential References Not Discussed:**

See suggested baseline selections above

**Experimental Designs Or Analyses:**

My biggest concern is in the authors' selection of baselines. For example, in section 4.1, where text-to-image models are aligned with prompt alignment and aesthetic quality, for inference-time technique, this work compares against best-of-n (BoN). This work chooses DPO and DDPO to represent finetuning-based methods. However, these baselines are not representative of the current literature on this topic, especially in their convergence speeds. In terms of fine-tuning based approaches, direct back-propagation-based methods such as DRAFT [1] and ELEGANT [2] are very effective in aligning diffusion models. And I am certain that tasks like prompt alignment and aesthetic quality permit employing these methods because the reward functions are differentiable.

On the other hand, BoN is also a pretty naive strategy. Why does this work not compare with more recent SMC-based methods? If the rationale is that these methods are all encapsulated by the 'general form' raised in this work, it may not be too convincing.

[1] https://arxiv.org/pdf/2309.17400
[2] https://arxiv.org/pdf/2402.15194

**Methods And Evaluation Criteria:**

Evaluation selection is appropriate.

**Other Comments Or Suggestions:**

In page 5, the authors discuss variants of intermediate reward functions. However, the results do not appear to be new. Many papers have proposed using (1) DDIM or (2) trained value functions to serve as intermediate rewards.

**Other Strengths And Weaknesses:**

1. I appreciate the writing flow of this work. It's easy to follow.

**Questions For Authors:**

1. Why did you choose simplistic inference-time methods (Best-of-N) and fine-tuning methods (DPO, DDPO) instead of more recent methods that are more representative? Please justify or include these additional comparisons.

2. Could you elaborate on the concrete novelty provided by the MAX and SUM potentials? How are these variants practically advantageous over the standard DIFFERENCE potential?

3. What exactly distinguishes your intermediate reward strategy from those already extensively explored in the literature using DDIM and trained value functions?

**Relation To Broader Scientific Literature:**

This work situates itself within recent efforts for inference-time steering of generative models, claiming to unify existing methods into a single theoretical framework

**Theoretical Claims:**

NA

---

> ### Author Rebuttal · Authors · 2025-04-01
>
> Thank you for your feedback on our paper, and for noting the clarity of our writing. We discuss your concerns below.
>
>
> _The main claim—unifying inference-time steering methods into a single FK-based framework— might lack sufficient novelty_
>
> -  We show that Feynman-Kac interacting particle systems, a well-known tool for rare-event sampling, are an effective and flexible framework for sampling from reward tilted diffusion models, where the reward can be any arbitrary scalar-valued function and the diffusion model can be continuous or discrete state.
>     -  In section F of the paper, we do show that prior works such as TDS and SVDD are specific instances of FK-IPS. However, we generalize beyond these choices and show the benefit of this generalization. We provide a detailed list of contributions below [[also see response to r3 below](https://openreview.net/forum?id=Jp988ELppQ&noteId=bBpOWtgFwj)].
>     -  Notably, these new choices for inference-time steering yield significant improvements.
>
> - **New choices yield higher-reward samples**. The product of potential condition we identify in equation 3 unlocks novel choices (of potentials, samplers, and rewards).
>     - *Potential choices*. For example, tables 3 & 5 show that new potential choices outperform traditional ones, such as the difference potential.
>     - *Intermediate rewards*. Furthermore, we show that diffusion models enable many choices for estimating intermediate rewards. In tables 4 & 5, we show that these new choices improve performance.
>
> _Why did you choose simplistic methods Best-of-N, DPO, DDPO?_
>
> - **Choice of DPO, DDPO**. We select DPO and DDPO as benchmarks as they provide public checkpoints and code, as opposed to DRAFT or ELEGANT. DPO is used in prior work, such as [Esser et al. 2024]. We are happy to add DRAFT or ELEGANT as benchmarks in the revised draft.
>
>
> - **FK steering can boost fine-tuning**. In Table 1, DPO does improve performance over the base model, but combining it with FK steering is even more performant. This indicates that fine-tuning and inference scaling are two complementary axes, and **any improvements provided by DRAFT or ELEGANT could be further increased by using FK steering.**
>
> - **Comparison to classifier-guidance**. We have also added results comparing FK steering to gradient guidance. FK steering outperforms both DDPO, DPO, best-of-n, and gradient guidance.
>
> | Model | GenEval | IR | HPS | Time  |
> |-|-|-|-|-|
> | SDv1.5                 | 0.44    | 0.187 | 0.245 | 2.4s  |
> | SDv1.5 + IR guidance   | 0.450    | 0.668 | 0.245 | 20s   |
> | SD v1.5 + FK (k = 4)   | **0.54**    | **0.898** | **0.263** | 8.1s  |
>
> - **Best-of-N is simple yet effective**. One exciting result in our paper is that best-of-n can even beat fine-tuning approaches (that require significant training compute) despite requiring no training.
>
> _Could you elaborate on the novelty provided by the MAX and SUM potentials?_
>
> - **Theoretical justification.** Based on eq. 3, many choices of potential provide a consistent approximation of the target distribution (see equation 3). However, when steering with arbitrary rewards, the DIFFERENCE potential does not necessarily yield high-reward samples.
>
>     - **Bounded rewards.**  Several choices of rewards are bounded. If a particle achieves maximum reward during sampling, the difference score will have to be negative and paradoxically will downweight this particle even though it has achieved a high reward. ImageReward is an example of such a bounded reward function.
>
> - **Empirical benefits.**  In tables 3 and 5, the MAX potential outperforms the DIFFERENCE and SUM potentials across model classes as well as different numbers of particles.
>
> _What distinguishes your intermediate rewards?_
>
> Prior works such as SVDD and TDS use the intermediate rewards $\log E_{p_\theta(x_0 \mid x_t)} \exp(r(x_0))$, either by approximating with the denoised expectation or by learning from model samples. In this work, we show that many choices of intermediate rewards can be used to consistently approximate the target distribution.
>
> 1. Unlike prior work that use intermediate rewards defined using model expectations, we show that the noise process and real data can be used to learn intermediate distribution via a regression objective.
> 2. Many-sample intermediate rewards: To the best of our knowledge, using samples from $p_\theta(x_0 | x_t)$ to define intermediate rewards $\log \sum_{i=1}^N \frac{1}{N} \exp(r(x_0^i))$ is a novel contribution of our work and was not proposed in prior works. The many-sample intermediate reward is a consistent estimator of $\log E_{p_\theta(x_0 | x_t)}\exp(r(x_0))$.
>
> ### References
> [Esser et al. 2024] Esser, Patrick, et al. "Scaling rectified flow transformers for high-resolution image synthesis." *Forty-first international conference on machine learning*. 2024.

---

### Official Review · Reviewer_xvFF · 2025-03-14

**Overall Recommendation:** 3

**Summary:**

The paper presents Feynman-Kac (FK) steering, a general particle-based framework for inference-time steering of diffusion models to generate outputs aligned with user-defined reward functions without requiring additional training or fine-tuning. FK steering generates multiple parallel sample trajectories from diffusion models and iteratively resamples these particles based on their scores computed using the potential function, which measures the user-interested property. Empirically, the method demonstrates substantial improvements in prompt fidelity and sample quality for both text-to-image and text diffusion models, outperforming fine-tuned baselines.

**Claims And Evidence:**

The paper claims that FK steering outperforms best-of-n sampling and gradient guidance, but there are some issues that need further clarification.

(1) Missing gradient guidance baseline in image experiments: While the paper includes gradient guidance comparisons in text diffusion tasks, it does not evaluate gradient guidance in image generation tasks, which use continuous-state diffusion models. Since gradient guidance is known to perform well in continuous diffusion settings, a direct comparison in image experiments would provide a fairer assessment of FK steering’s advantages.

(2) FK steering involves additional computation per particle due to resampling steps, making a direct comparison to best-of-n with the same number of particles k somewhat unfair. Since FK steering does not significantly outperform best-of-n in Table 1 and 2 (especially Table 1), this raises questions about its computational efficiency and whether its performance gains justify the added cost. Including more balanced comparisons that account for computational overhead would strengthen the evaluation.

**Essential References Not Discussed:**

For gradient guidance, the paper should cite "Diffusion Posterior Sampling for General Noisy Inverse Problems, Hyungjin Chung, Jeongsol Kim, Michael T. McCann, Marc L. Klasky, and Jong Chul Ye", which originally proposed the concept of training-free gradient guidance.

**Experimental Designs Or Analyses:**

As mentioned before, the lack of gradient guidance baselines in image tasks, unfair compute comparison with best-of-n, missing diversity analysis, and limited resampling strategy evaluation weaken the rigor of the results. Addressing these would improve fairness and clarity.

**Methods And Evaluation Criteria:**

Selecting the best sample from multiple trajectories can reduce the diversity of generated outputs. Since diversity is an important factor in many generative tasks, incorporating it as an additional evaluation metric would provide a more comprehensive assessment of FK steering’s effectiveness and potential trade-offs.

**Other Comments Or Suggestions:**

NA

**Other Strengths And Weaknesses:**

The proposed framework is easy to implement, plug-and-play, and gradient-free, making it highly practical for real-world applications. Additionally, the paper is well-written and easy to follow, with clear explanations that effectively communicate the method and its contributions.

**Questions For Authors:**

Please refer to the comments above. I am open to increasing the rating if all concerns are adequately addressed.

**Relation To Broader Scientific Literature:**

The paper integrates ideas from diffusion model guidance, sequential Monte Carlo, rare-event sampling, and reinforcement learning-based fine-tuning, offering a unified inference-time steering framework. By demonstrating that FK steering can match or outperform fine-tuning approaches with lower computational cost, it contributes to the growing trend of efficient and controllable generative modeling. However, a direct comparison to recent adaptive guidance techniques and a more thorough analysis of computational efficiency trade-offs would further contextualize its impact within the broader literature.

**Theoretical Claims:**

The paper's theoretical claims, particularly the consistency of FK steering’s particle-based approximation to the target distribution, are well-grounded in sequential Monte Carlo theory and FK interacting particle systems.

---

> ### Author Rebuttal · Authors · 2025-04-01
>
> Thank you for your thoughtful comments, and for highlighting that FK steering is a practical, effective approach for efficient, controllable generation with diffusion models. We also appreciate your assessment of our paper's clarity and strong theoretical grounding.
>
> We address each of your concerns below. Let us know if there is any additional information we can include that would be helpful for your review.
>
> _Missing gradient guidance baseline in image experiments._
>
> 1. **Requested gradient guidance results**.
>     - FK steering (with 4 particles, gradient-free) **outperforms gradient guidance** on all metrics and **is significantly faster** (8.1 vs 20 seconds for SDv1.5)!
>     - FK steering combined with ImageReward gradient guidance performs even better but requires a significant increase in sampling time.
>
> ### Gradient-guidance Results
> | Model | GenEval | IR | HPS   | Sampling Time  |
> |-|-|-|-|-|
> | SDv1.5   | 0.44    | 0.187 | 0.245 | 2.4s  |
> | SDv1.5 + IR guidance   | 0.450    | 0.668 | 0.245 | 20s   |
> | SD v1.5 + FK (k = 4)   | **0.54**    | **0.898** | **0.263** | 8.1s  |
> | SD v1.5 + IR guidance + FK (k = 4)   | **0.56**    | **1.290** | **0.268** | 55s |
>
> Thank you for encouraging us to run these experiments. These results strengthen our paper and we will include them in our final draft.
>
> 2. **Additional advantages over gradient guidance**. Gradient-free steering has the following additional benefits:
>     - **Enables steering with non-differentiable rewards**. Gradient-free steering methods enable the use of non-differentiable rewards such as perplexity from an autoregressive model or a trigram model, see table 2 in the paper for examples. Additionally, gradient-free steering can make use of closed-source reward models hosted via APIs, or non-differentiable constraints.
>     - **Enables steering discrete models**. Gradient-free steering of masked diffusion models can enable attribute control (tables 2 and 4). More recently, LLaDA [Nie et al 2025] showed that discrete-state text diffusion models are competitive with auto-regressive models, making the development of gradient-free steering even more pertinent.
>
> _FK steering involves additional computation per particle due to resampling steps...this raises questions about its computational efficiency and whether its performance gains justify the added cost._
>
> 1.  **Computational  efficiency**. This is an excellent question. In the text-to-image generation experiments, using interval re-sampling (calling the reward model every 20 of 100 total steps) results in minimal compute overhead. We include timing results in the appendix (table 6) which show that FK steering results in a **minimal increase in sampling time compared to best-of-n**. For our most performant model, SDXL, the increase in sampling time is only 3%, see table below. We also note that as model size increases, the gap between FK and best-of-n reduces further.
>
> | Model         | Params |   BoN (*k = 4*) | FK (*k = 4*) |
> |-|-|-|-|
> | SDXL          | 2.6B             | 42.3s          | 43.5s        | 21.7s                  |
>
> 2. **Improved results justify computational overhead**. In our controllable text generation experiments (see table 4) with the masked diffusion language model, FK steering with **k=4** achieves attribute accuracy of **22.0%**, significantly outperforming the **best-of-8** attribute accuracy of **3.7%**. Best-of-8 takes 19.3 seconds versus 20.7 seconds for FK steering(k=4) on a single A100 GPU. We have added the timing results to the revised draft.
>
> _Diversity is an important factor...incorporating it as an additional evaluation metric would provide a more comprehensive assessment of FK steering’s effectiveness and potential trade-offs._
>
> - **Diversity Results**. Currently, we include CLIP-diversity scores in table 7, and provide 4 and 8 particle run samples, and highlight different components of the algorithm that can affect diversity. In the revised draft, we will include global diversity metrics as well.
>
> _...a direct comparison to recent adaptive guidance techniques and a more thorough analysis of computational efficiency trade-offs would further contextualize its impact within the broader literature._
>
>  - **Adaptive guidance techniques**. Other forms of conditioning, beyond the classifier and classifier-free guidance approaches we evaluate, could be used for the proposal generator in FK steering. We are very interested in exploring building on top of other recent methods in future work.
>
>  - **Computational efficiency trade-offs**. In table 6, we include the sampling time of FK steering compared to best-of-n. We discuss additional timing results above.
>
> _For gradient guidance, the paper should cite "Diffusion Posterior Sampling for General Noisy Inverse Problems"_
>
> Thank you. We will update our related works section to include this work in our final draft.
>
>
> ### References
> Nie et al. "Large Language Diffusion Models." arXiv preprint arXiv:2502.09992 (2025).

---

### Decision · Program_Chairs · 2025-05-01

**Decision:**

Accept (poster)

**Comment:**

This paper proposes a framework for steering diffusion models without retraining them. The method is based on the Feynman-Kac (FK) steering. Multiple trajectories are generated from a diffusion model and iteratively resampled based on a score provided by a potential function that measures how well the user requested property is being satisfied. Empirically, the method is shown to improve prompt fidelity and sample quality for both text-to-image and text diffusion models, outperforming fine-tuned baselines.

By and large reviewers recognized the clarity of the manuscript and the experimental evidence.  Reviewers have raised some concerns about novelty and comparison with other baselines.

During the rebuttal the authors have provided more empirical evidence and comparisons with the requested baselines and clarified
some of the phrasing regarding the novelty of the work which resulted in some scores being increased.